# Identifying essential factors for energy-efficient walking control across a wide range of velocities in reflex-based musculoskeletal systems

**Shunsuke Koseki**⬤*, **Mitsuhiro Hayashibe, Dai Owaki**

Department of Robotics, Graduate School of Engineering, Tohoku University, Sendai, Japan

* shunsuke.koseki.q4@dc.tohoku.ac.jp

## Abstract

Humans can generate and sustain a wide range of walking velocities while optimizing their energy efficiency. Understanding the intricate mechanisms governing human walking will contribute to the engineering applications such as energy-efficient biped robots and walking assistive devices. Reflex-based control mechanisms, which generate motor patterns in response to sensory feedback, have shown promise in generating human-like walking in musculoskeletal models. However, the precise regulation of velocity remains a major challenge. This limitation makes it difficult to identify the essential reflex circuits for energy-efficient walking. To explore the reflex control mechanism and gain a better understanding of its energy-efficient maintenance mechanism, we extend the reflex-based control system to enable controlled walking velocities based on target speeds. We developed a novel performance-weighted least squares (PWLS) method to design a parameter modulator that optimizes walking efficiency while maintaining target velocity for the reflex-based bipedal system. We have successfully generated walking gaits from 0.7 to 1.6 m/s in a two-dimensional musculoskeletal model based on an input target velocity in the simulation environment. Our detailed analysis of the parameter modulator in a reflex-based system revealed two key reflex circuits that have a significant impact on energy efficiency. Furthermore, this finding was confirmed to be not influenced by setting parameters, i.e., leg length, sensory time delay, and weight coefficients in the objective cost function. These findings provide a powerful tool for exploring the neural bases of locomotion control while shedding light on the intricate mechanisms underlying human walking and hold significant potential for practical engineering applications.

## Author summary

Previous reflex-based control systems face significant limitations in accurately regulating walking velocity owing to the vast number of control parameters involved. This hinders identifying essential reflex circuits that have a significant impact on energy-efficient

**Data Availability Statement:** All experimental data and programming code are publicly available on GitHub, at https://github.com/Shunsuke-KK/reflex_plos_revision.

**Funding:** This work was supported by the JSPS KAKENHI Grant-in-Aid for Scientific Research on Innovative Areas Hyper-Adaptability Project (JP20H05458 and JP22H04764, M.H.) and Scientific Research (A) (JP23H00481, D.O.). The funders had no role in study design, data collection and analysis, decision to publish, or preparation of the manuscript.

**Competing interests:** The authors have declared that no competing interests exist.

walking across a wide range of speeds. Our research tackles this challenge by developing a reflex-based control framework that precisely regulates the velocities of the bipedal model through the performance-weighted least squares (PWLS) method that optimizes the control parameter values while considering energy efficiency. Through a detailed analysis, we identify two key reflex circuits essential for generating energy-efficient walking over a wide range of velocities. We hope that our research will inspire future investigations into reflex mechanisms and facilitate the development of advanced walking control systems for practical applications, such as gait-assisted exoskeletons and prosthetic legs, and robot control.

## Introduction

Walking is a fundamental mode of locomotion in our daily lives. The neurological and biochemical control mechanisms that support it comprise one of the most complex autonomous control systems in the human body [1–6]. Modeling and replicating the underlying walking mechanisms is expected to contribute to engineering applications including energy-efficient bipedal robots [7–11], gait-assisted exoskeletons [12–15], smart prosthetic legs [16, 17].

One of the most crucial aspects of human walking is the energy-efficient maintenance of our controlled velocity in the range of 1.0–1.6 m/s [18]. Central pattern generators in the central nervous system engage rhythmic neural circuits that generate basic leg movements [19–23]. Equally indispensable are reflex control mechanisms, which provide rapid adjustments to external stimuli from the sensory organs [19–21, 24], even for unexpected balance loss or unpredicted ground variations during walking, thereby maintaining velocity. Notably, humans modulate the reflex response depending on the task in locomotion [25, 26]. These findings indicate that reflex mechanisms have a crucial impact on achieving stable and energy-efficient walking.

A physical simulation is a powerful tool for exploring the neural basis of walking, a complex phenomenon generated by the interaction of versatile mechanisms [27]. Previous studies have shown that reflex-based control systems can generate human-like walking in terms of muscle activities, joint angles, and torque patterns [28–37]. Notwithstanding the great advancements thereof, these systems have difficulty regulating velocity, owing to the large number of control parameters that must be properly tuned [38], e.g., Geyer [28] used 36 control parameters to generate a steady gait in a two-dimensional musculoskeletal model. Previous studies have attempted to regulate velocities within the reflex-based control frameworks [30, 32], but their methods were limited to transitions between predetermined velocities and did not provide precise velocity controls. Furthermore, the transitions between these predetermined velocities were accomplished through the utilization of distinct control parameters designated for these transitions. Given that walking speed assumes continuous values, achieving precise control over walking speed through such means would theoretically require infinite parameters for transitions. Because "transition" differs from "control" in terms of its adaptability, it is essential to extend reflex-based systems to affect precise velocity controls while improving energy efficiency to explore reflex control mechanisms in walking. Furthermore, the development of energy-efficient control over a wide range of velocities in the reflex-based system will lead to improving the performance of the controller for exoskeletons [39, 40] and prosthetic leg [17, 41] by adjusting their control parameters according to user walking velocity.

The purpose of this study is to extend the reflex-based control system to enable controlled walking velocities based on target speeds and gain a better understanding of its modulation

and energy-efficient maintenance mechanism across a wide range of walking velocities. To achieve this, we developed a novel performance-weighted least squares (PWLS) method to design a parameter modulator that coordinates a vast number of control parameters for an input target velocity while maintaining energy efficiency. In short, the reflex-based control system with the parameter modulator optimized via the PWLS successfully and energetic-efficiently maintains the desired velocity from 0.7 to 1.6 m/s in a two-dimensional musculoskeletal model. Subsequently, the detailed analysis of the parameter modulator in the reflex-based system identified two key reflex circuits affecting energy efficiency across a wide range of walking velocities. The contributions of this work include (i) extending a reflex-based control system to include velocity control, (ii) providing an adaptive polynomial regression method that uses performance indices to control performance, and (iii) identifying the key reflex circuits related to energy efficiency across a wide range of walking velocities.

## Generating walking through reflex-based control

### Musculoskeletal model

A two-dimensional musculoskeletal model with a height of 180 cm and weight of 80 kg was employed in this study, as illustrated in Fig 1A. Detailed parameters are listed in Table 1. The model was constructed within a MuJoCo simulation environment [42]. We employed Mujoco because of its ability to conduct simulations at a relatively low computational cost while ensuring the validity of the physical calculations. Many physics engines such as OpenSim, which is often used in biomechanics research, models contact with the ground by using a spring-damper model. This approach requires using smaller timesteps to prevent the model foot from penetrating the ground, which increases computational cost. MuJoCo introduces several formulations of the physics of contact. This allows for more efficient calculations and versatile contact behaviors. As recent studies [43, 44] have aimed to transfer OpenSim models to MuJoCo to improve computational speed and use versatile contact behaviors, MuJoCo is an acceptable physics engine with physical validity involving contact. The model consists of 12 segments: torso, hip, two thighs, two knees, two shanks, two feet, and two sets of toes, based on previous studies [28, 29]. The motions of the model are constrained to the sagittal plane. The hip and knee segments possess no mass. The model has nine internal degrees of freedom, as illustrated in Fig 1B; a torso joint, hip joints, knee joints, ankle joints, and toe joints. The spring and damper constants for all joints apart from the toe joints are set to 0 Nm/rad and 1 Nms/rad, respectively. The toe joints are set with a spring constant of 30 Nm/rad, which is consistent with [29], and a damping constant of 0 Nms/rad. The spring and damper within the joint generate a torque that is proportional to the distance from the equilibrium position and friction that is proportional to the joint's angular velocity, respectively. We set the sliding friction coefficient between the feet and the ground at a relatively high value, 2.5. Then, sliding is unlikely to happen. Fig 1A illustrates the locations of the muscle actuators that generate torque in the joints. Each leg has eight muscle actuators reflecting the gluteal (GLU), hip flexor (HFL), vasti (VAS), tibialis anterior (TA), soleus (SOL), hamstring (HAM), rectus femoris (RF), and gastrocnemius (GAS) musculature.

### Muscle actuator model

The musculoskeletal model includes actuators that simulate biological muscle actions that produce torque in the joints [42, 45]. The muscle actuator model comprises an inelastic tendon with a rest length of $l^0$ and a muscle that contracts the tendon. The muscle actuator force, $F$, is a function of muscle length $l$, velocity $v$, and current activation level $a \in [0, 1.0]$. A higher activation level results in a greater force being produced by the muscle. For the computation, $l$ and

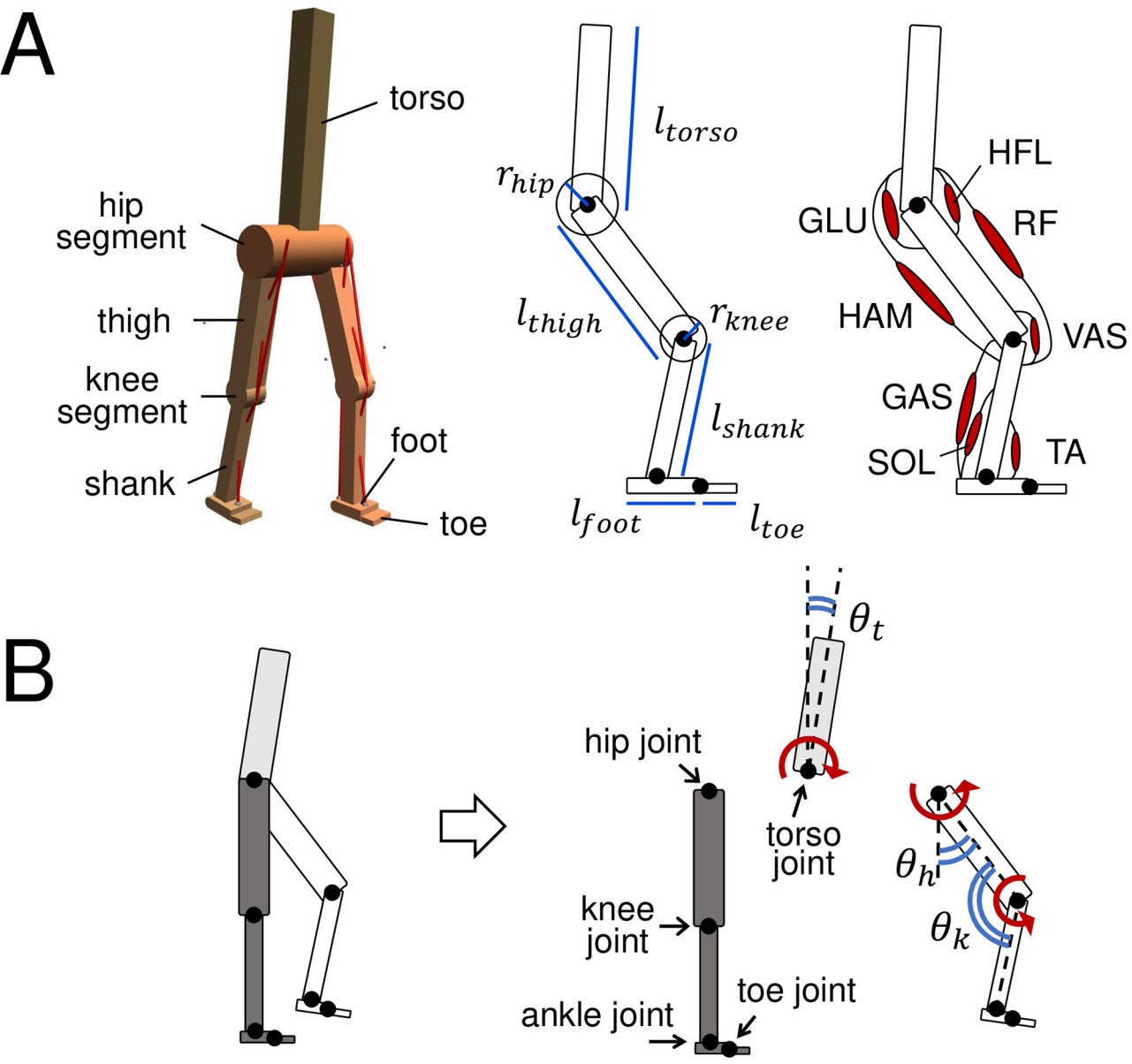

**Fig 1. Musculoskeletal model employed in this study. A. Structure of the musculoskeletal model.** Left: Oblique view of the bipedal model in the MuJoCo simulator. Center: Side view of the model with segment parameters. Right: Muscle alignments. **B. Internal degrees of freedom of the model.** $\theta_t$, $\theta_h$, and $\theta_k$ represent torso joint angle, hip joint angle, and knee joint angle, respectively. The red arrows depict the positive direction of joint rotation.

$v$ are scaled by the equilibrium length, $l^0$, as follows:

$$\tilde{l} = \frac{l}{l^0}, \tag{1}$$

$$\tilde{v} = \frac{v}{l^0}. \tag{2}$$

**Table 1. Musculoskeletal model parameters.**

| Parameter | Value | Parameter | Value |
|---|---:|---|---:|
| $m_{torso}$ | 53.5 kg | $l_{torso}$ | 80 cm |
| $m_{hip}$ | 0 kg | $r_{hip}$ | 9 cm |
| $m_{thigh}$ | 8.5 kg | $l_{thigh}$ | 50 cm |
| $m_{knee}$ | 0 kg | $r_{knee}$ | 5.5 cm |
| $m_{shank}$ | 3.5 kg | $l_{shank}$ | 40 cm |
| $m_{foot}$ | 1.0 kg | $l_{foot}$ | 15 cm |
| $m_{toe}$ | 0.25 kg | $l_{toe}$ | 5.25 cm |

The actuator force, $F$, is computed as follows:

$$F(\tilde{l}, \tilde{v}, a) = \{aF_l(\tilde{l})F_v(\tilde{v}) + F_p(\tilde{l})\} \cdot F^0, \tag{3}$$

where $F_l$ and $F_v$ represent the force–length and force–velocity relationships, respectively. $F_p$ represents the passive force that is always present, regardless of activation, and $F^0$ denotes the maximum isometric force that takes different values for each muscle actuator, as listed in Table 2. These values are consistent with those of previous studies [28, 29]. The computations of $F_l$, $F_v$, and $F_p$ are detailed in S1 Appendix. Briefly, $F_l$ is a function that attains a maximum value at $l = l^0$, $F_v$ is a function that returns a smaller value for faster contraction of the muscle actuator, and $F_p$ increases monotonically for $\tilde{l}$. The muscle current activation level, $a$, is calculated for the input stimulation signal, $u \in [0, 1.0]$:

$$\frac{\partial a}{\partial t} = \frac{u - a}{\tau(u, a)}, \tag{4}$$

where

$$\tau(u, a) = \begin{cases} 0.01 \cdot (0.5 + 1.5a) & (u > a) \\ 0.04/(0.5 + 1.5a) & (u \leq a) \end{cases}, \tag{5}$$

## Reflex-based control

The reflex-based controller employed in this study is identical in principle to that of Wang et al [29]. This controller is based on the Geyer model and incorporates additional control laws to adjust the target hip joint angle and to fix the hip and knee joints prepared for the heel strike. Consequently, it more robustly produces gaits and can manage non-steady states during transitions in walking velocities. Moreover, the extended model can maintain the biomechanical explanatory ability essential to discussing gait adaptation mechanisms, which may be lost through simplification. We did not make any ad-hoc adjustments to the basic reflex-based control model to ensure optimal simulation performance. The controller computes the muscle stimulation, denoted as $u_i$, for each muscle, $i$, by incorporating sensory feedback with a time

**Table 2. Individual $F^0$.**

| | SOL | TA | GAS | VAS | HAM | RF | GLU | HFL |
|---|---|---|---|---|---|---|---|---|
| $F^0$[N] | 4000 | 800 | 1500 | 6000 | 3000 | 1000 | 1500 | 2000 |

delay of $\Delta t$ as input. The control law switches depending on whether the leg is in the stance or swing phase. Moreover, additional stimulation is introduced during the late stance and late swing phases. The control system comprises three primary functions: force feedback, length feedback, and muscle-driven proportional derivative (PD) control.

- Force feedback:
  The force feedback law returns the stimulation signal, $u_i^F$, in response to the actuator force, $F_i$. In humans, the signals of the $F_i$ arise from Golgi tendon organs and are carried by type Ib afferents to the spinal cord. $u_i^F$ is calculated as follows:

$$u_i^F = G_i \tilde{F}_i(t - \Delta t_i), \tag{6}$$

$$\tilde{F}_i(t - \Delta t_i) = \frac{F_i(t - \Delta t_i)}{F_i^0}, \tag{7}$$

  where the gain, $G_i$, is the positive control parameter, and $\tilde{F}_i(t - \Delta t_i)$ is the scaled actuator force, (i.e., $\tilde{F}_i = F_i/F_i^0$), which includes the sensory time delay, $\Delta t_i$.

- Length feedback
  Through length feedback, we calculate the stimulation signal, denoted as $u_i^l$, corresponding to the length of the muscle actuator, $l_i$. This function models the stretch reflex of the muscle spindle. $u_i^l$ is calculated as follows:

$$u_i^l = \max\{0, G_i(l_i(t - \Delta t_i) - l_i^{tar})\}, \tag{8}$$

  where the target length, $l_i^{tar}$, and gain, $G_i$, are positive control parameters.

- Muscle-driven PD control
  The muscle-driven PD control generates the stimulation necessary to move joint $\theta_j$ (Fig 1B) to the target angle, $\theta_j^{tar}$. This function can be interpreted as a polysynaptic reflex that is mediated by the joint position afferent input from group III fibers and descending signals of the target joint angles from the supraspinal. The muscle-driven PD control laws are employed to stabilize the generated gait by the hip muscles during the stance phase to balance the torso and during stance preparation to prepare for ground contact (described in S1 Appendix).
  For muscle actuator, $i$, which applies torque to joint $\theta_j$ in the positive direction, $u_i^{\theta_j}$ is defined as follows:

$$u_i^{\theta_j} = \max\{0, K_i(\theta_j^{tar} - \theta_j(t - \Delta t_i)) - D_i \dot{\theta}_j(t - \Delta t_i)\}. \tag{9}$$

  Conversely, for muscle actuator $i$, which applies torque in the negative direction, $u_i^{\theta}$ is defined as follows:

$$u_i^{\theta} = \max\{0, K_i(\theta_j(t - \Delta t_i) - \theta_j^{tar}) + D_i \dot{\theta}_j(t - \Delta t_i)\}, \tag{10}$$

  where the proportional gain, $K_i$, and derivative gain, $D_i$, are positive control parameters.

Fig 2 illustrates the reflexes in the stance phase and swing phase. Please see S1 Appendix or the original paper [29] for more details.

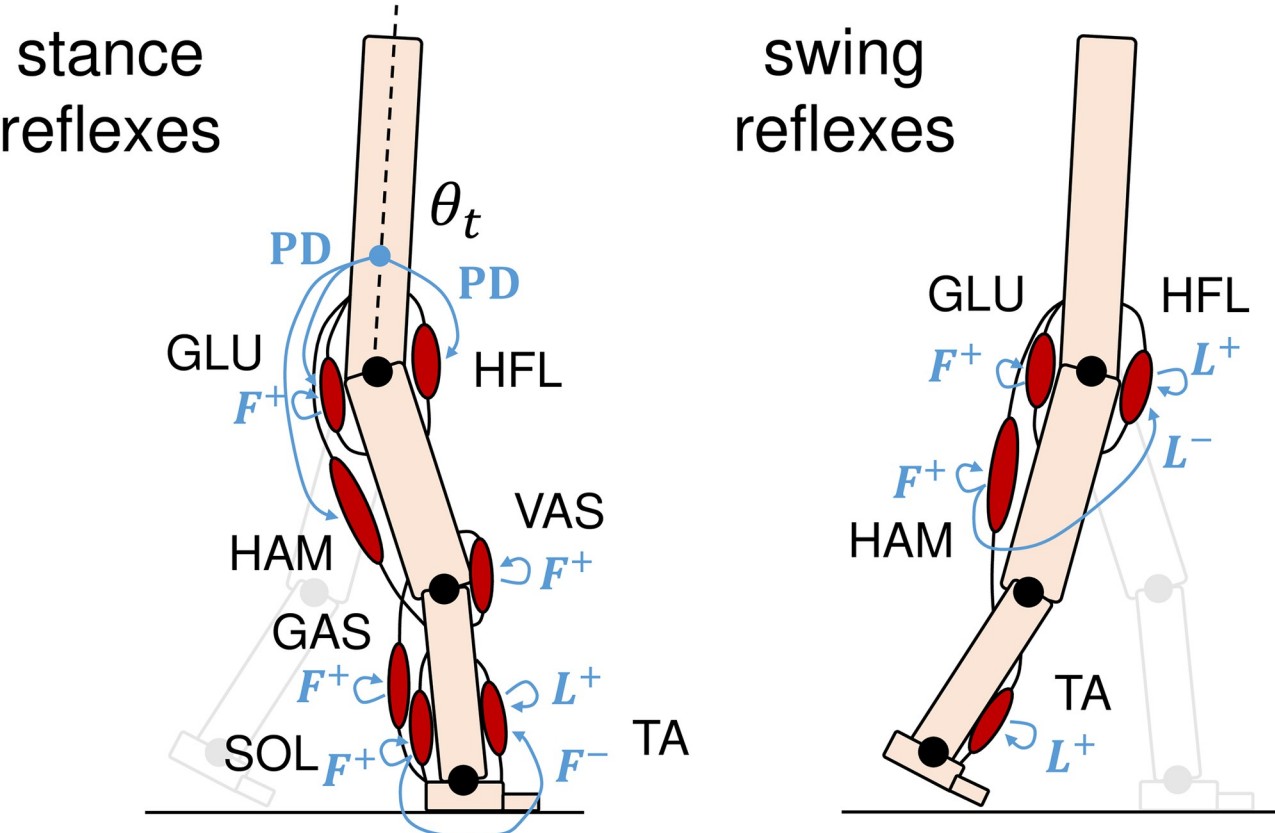

**Fig 2. Key reflexes in the stance and swing phase.** *F* represents force feedback, *L* represents length feedback, and PD represents muscle-driven PD control. + and − denote positive and negative feedback, respectively. In the stance phase, $F^+$ at GLU, VAS, and SOL generate compliant leg behavior. $L^+$ at TA prevents overextension of the ankle joint, which is suppressed by the $F^-$ from the SOL. $F^+$ at GAS contributes to push-off and prevents overextension of the knee joint. Muscle-driven PD controls at HFL, GLU, and HAM balance the torso. In the swing phase, $L^+$ facilitates leg swing, which is suppressed by $L^-$ from HAM. $F^+$ at GLU and HAM apply braking force to the swing leg. $L^+$ at TA raises the toes to create clearance between the feet and the ground.

## Modulation of control parameters

Fig 3 displays the control diagram of the musculoskeletal model, which is structured into spinal and supraspinal layers. We added a parameter modulator that allows a wide range of walking while maintaining energy efficiency to the previous model [29]. The reflex-based controller in the spinal layer activates muscles. The parameter modulator in the supraspinal layer coordinates the control parameter set $Y^{tar}$ to achieve the input target velocity, $v_{vel}^{tar}$. Each control parameter, $y_i^{tar} \in Y^{tar}$, is calculated using the function $P_i(v_x)$ in the parameter modulator for the input target velocity, $v_x^{tar}$ (Fig 4). $P_i(v_x)$ is derived via the polynomial regression of the relationship between the velocity and parameter value, which is collected using optimization by incrementally increasing/decreasing the target velocity. SIMBICON in the supraspinal cord adjusts the desired foot placements.

## Optimizing control parameters

The dataset for polynomial regression is collected via optimization by incrementally changing the target velocity. In total, there are 56 control parameters, $Y \in \mathbb{R}^{56}$. Details are provided in S1 Appendix. We optimize these parameters using the covariance matrix adaptation evolution

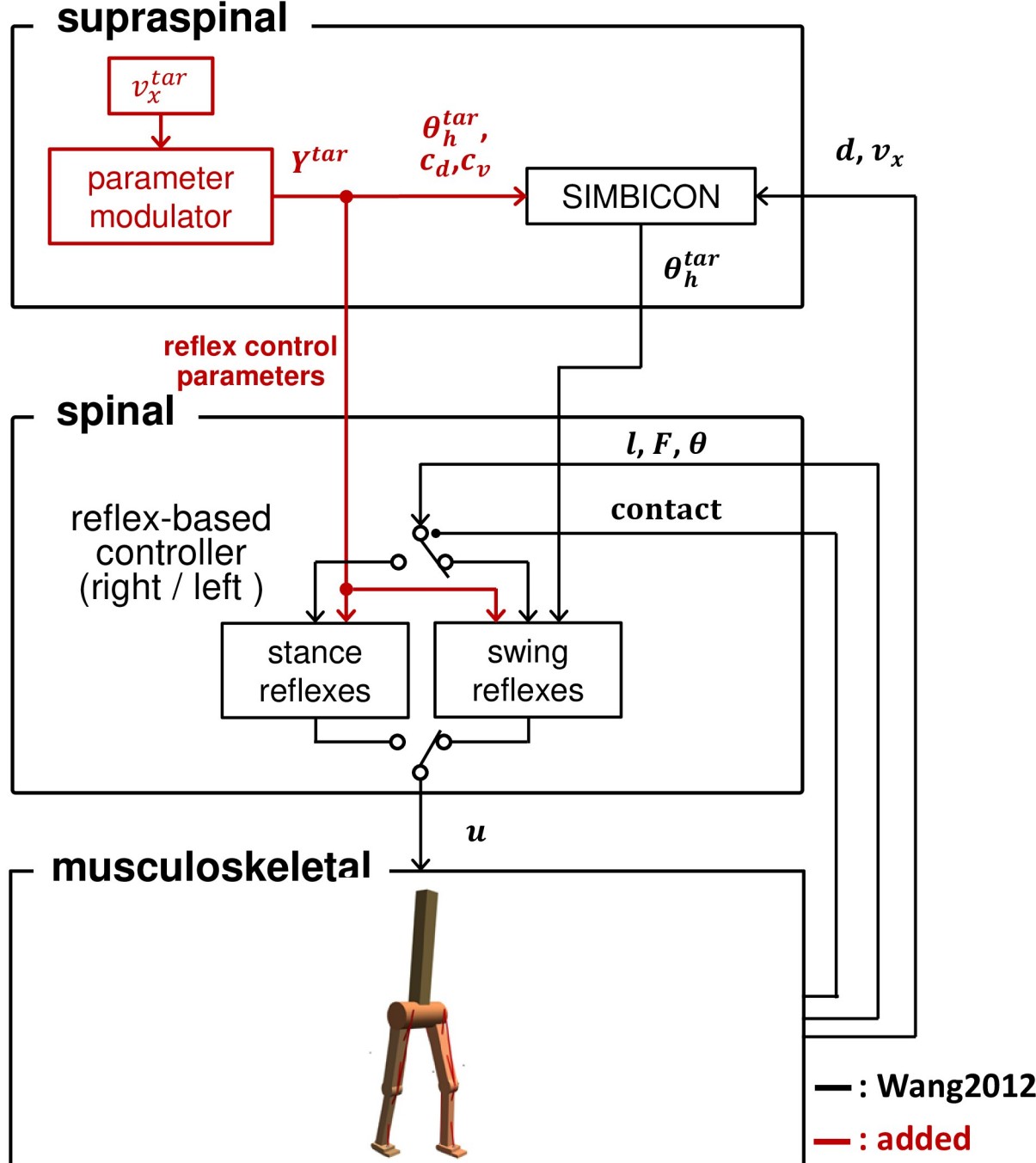

**Fig 3. Control diagram for the musculoskeletal model.** The musculoskeletal model is driven by activated muscles receiving stimulation signals, *u*, from the reflex-based controller in the spinal layer. Each leg (right and left) is controlled by separate stance and swing reflexes. The control law switches depending on whether the leg is in contact with the ground. Sensory information from the model is fed back to the controller for motion generation. The supraspinal layer controls the walking velocity and adjusts the desired foot placements. The parameter modulator coordinates the control parameter set $Y^{tar}$ to achieve input target velocity, $v_{vel}^{tar}$. SIMBICON adjusts the target hip angle, $\theta_h^{tar}$, in the stance preparation phase using the sensory information (*d* and $v_x$, see S1 Appendix).

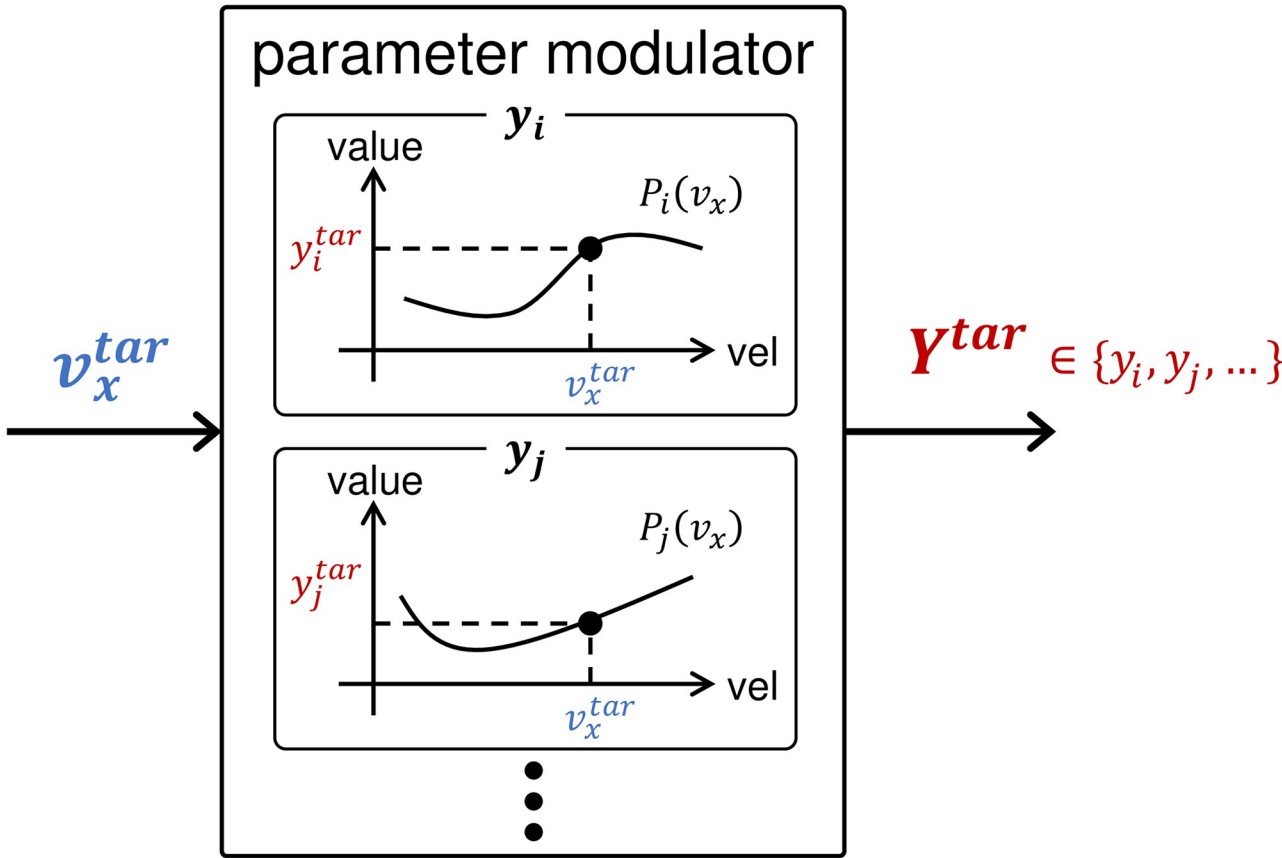

**Fig 4. Parameter modulator.** This is located in the supraspinal layer of the controller (Fig 3). By using the polynomial function, $P_i$, each control parameter, $y_i$, is adjusted to achieve the input target velocity, $v_x^{tar}$.

strategy (CMA-ES) algorithm [46], which is well-suited for nonlinear and non-convex optimization problems. In this algorithm, independent $\lambda$ search points are sampled from a multivariate normal distribution, $\mathcal{N}$, at each generation, $g$:

$$Y^{(g+1)} \sim \mathcal{N}(\boldsymbol{m}^{(g)}, \sigma^{(g)^2} \boldsymbol{C}^{(g)}), \tag{11}$$

where $Y$ represents the reflex control parameter set in this study, $\boldsymbol{m}^{(g)}$ represents the mean value of the search distribution at generation $g$, $\sigma^{(g)}$ represents the step size at generation $g$, and $\boldsymbol{C}^{(g)}$ represents the covariance matrix at generation $g$. The sampled control parameter set, $Y$, is evaluated using the objective function, $f$. Then, using the top $\mu$ data points from $\lambda$ offspring along with the evolution paths, $p_\sigma^{(g)}$ and $p_C^{(g)}$, which accumulate historical search directions, $\boldsymbol{m}$, $\sigma$, $\boldsymbol{C}$, $p_\sigma$, and $p_C$ are updated.

The evaluation of $Y$ is conducted from a fixed initial state until time step $T'$, in which the musculoskeletal model falls, with an upper limit $T$. More specifically, if the model falls before reaching an upper time step $T$, the evaluation is terminated at that time step $T'$, and else $T' = T$. We judge that the model has fallen when any segment above the knee comes into contact with the ground. By terminating the unnecessary evaluations prior to reaching the maximum time step, the computation time can be reduced. Optimization is conducted to determine the optimal solution that minimizes the objective cost function. To generate an energy-efficient

gait that follows the target velocity, the objective cost function, $f$, is designed as follows based on previous studies [29, 30]:

$$f(\boldsymbol{Y}) = \sum_{t=1}^{T'} r(\boldsymbol{s}_t) + \alpha_E(\text{CoT} - 0.3), \tag{12}$$

where $\boldsymbol{s}_t$ represents the state of the model at time step $t$ as determined by the control parameter set, $\boldsymbol{Y}$, $r$ represents the reward function for state $\boldsymbol{s}_t$, $\alpha_E$ represents the weight coefficient, and CoT represents the cost of transport [47]. Reward function $r$ is defined as follows:

$$r(\boldsymbol{s}_t) \quad = r_{alive} + r_{forward} + r_{torso} + r_{fall}, \tag{13}$$

$$r_{alive} \quad = -1, \tag{14}$$

$$r_{forward} \quad = \min(1, \alpha_v |v_x - v_x^{tar}|^2), \tag{15}$$

$$r_{torso} \quad = \alpha_t \theta_t^2, \tag{16}$$

$$r_{fall} \quad = \begin{cases} 0 \\ 5,000 \quad (\text{fall down} \wedge T' \le 700) \end{cases}, \tag{17}$$

where $r_{alive}$ prevents the model from falling. If the model does not fall over long timesteps, this term can reduce the total cost. $r_{forward}$ represents the penalty for the difference between the current horizontal walking speed, $v_x$, and the target velocity, $v_x^{tar}$, with a lower limit of -1. $r_{torso}$ is established to maintain an upright torso. $r_{fall}$ incurs a large cost if the model falls within 700 timesteps of the initial state. This prevents convergences at local minima, where the model falls forward immediately from the initial position at the target velocity. $\alpha_E(\text{CoT} - 0.3)$ in $f$ represents the energy cost added at the end of a trial. 0.3 is the deviation, which in turn permits a larger weight coefficient on CoT. The CoT quantifies the energy efficiency of locomotion, and a lower value indicates better energy efficiency [47]. The CoT is expressed by the following equation:

$$\text{CoT} = \frac{J}{Mg\Delta d}, \tag{18}$$

where $J$ represents the total metabolic energy, $M$ represents the model mass, $g$ represents the acceleration of gravity, and $\Delta d$ represents the distance traveled. $J$ is calculated by summing the total metabolic energy expended by all muscles, as described in previous studies [29, 48]. The detailed equations for $J$ are provided in S1 Appendix.

## Dataset collection for polynomial regression

We run two programs in parallel to collect the data efficiently. Within one thread, initially, the target velocity, $v_x^{tar}$, in cost function $f$ (Eq (12)) is set to 1.3 m/s, which is the human self-selected speed [29]. Control parameter set, $\boldsymbol{Y}$, that generate a gait around $v_x^{tar}$ are obtained. Then, $v_x^{tar}$ is slightly increased to $v_x^{tar} + \Delta v_x$, and the corresponding control parameter set around the updated target velocity are collected. This process is repeated until $v_x^{tar}$ reaches the upper limit of the target velocity, $v_{x\,max}^{tar}$ (See S1 Appendix in detail). Within the other thread, the target velocity, $v_x^{tar}$ is initially set to 1.2 m/s. Then, $v_x^{tar}$ is slightly decreased to $v_x^{tar} - \Delta v_x$ and this process is also repeated until $v_x^{tar}$ reaches the lower limit of the target velocity, $v_{x\,min}^{tar}$. The initial control parameters are set identically in both programs.

The control parameters, $Y$, are optimized using the $G$ generation for each target velocity, $v_x^{tar}$. When $v_x^{tar}$ is changed, $\sigma$ in Eq (11) is reset to $\sigma^0$ and $p_\sigma$, and the $p_C$ are emptied, whereas $m$ and $C$ are maintained. If model-walking is maintained at the upper timestep limit of the evaluation trial with control parameters $Y$, a tuple consisting of walking speed, control parameters, and CoT $\{(v_x, Y, CoT)\}$ is added to the dataset. Notably, the dataset with $n$ data points, $\{(v_{x1}, Y_1, CoT_1), \ldots, (v_{xn}, Y_n, CoT_n)\}$, is sorted according to velocity $v_x$ (i.e., $v_{x1} \leq v_{x2} \leq \ldots \leq v_{xn}$).

## PWLS (Performance Weighted Least Square method)

Each control parameter, $y_i^{tar} \in Y^{tar}$, is modulated through an $m$th degree polynomial function, $P_i(v_x)$ (Fig 4), for the input velocity, $v_x$, as follows:

$$y_i = P_i(v_x), \tag{19}$$

$$= \omega_{i0} v_x^0 + \omega_{i1} v_x^1 + \ldots + \omega_{im} v_x^m, \tag{20}$$

where $\omega_{ij}$ are coefficients calculated using our proposed PWLS, a polynomial regression algorithm that minimizes the total squared performance-weighted error of data points. Unlike the normal least-squares method in which data points are treated without bias when calculating the total squared error, PWLS derives the polynomial function from dataset with bias. Thus, it assigns a greater weight to higher-performing data points to reinforce energy-efficient walking via regression. PWLS is similar to the weighted least-squares method [49]. The weighted least square method is the polynomial regression algorithm that is used when handling heteroscedastic data, meaning that the variance among the measured points is not constant. The variables that cause the variance of observations are incorporated to weigh each data point. In our PWLS, each data point is weighted according to its performance instead of the factor that causes heteroscedastic.

Suppose there are $n$ data points, $\{(v_{x1}, y_{i1}), \ldots, (v_{xn}, y_{in})\}$, for each control parameter from the collected dataset, $\{(v_{x1}, Y_1, CoT_1), \ldots, (v_{xn}, Y_n, CoT_n)\}$. The total error between $y_i$ and the expected value derived from $P_i$ can be expressed in matrix form as

$$\| y_i - V\omega_i \|_2 = \left\| \begin{bmatrix} y_{i1} \\ y_{i2} \\ \vdots \\ y_{in} \end{bmatrix} - \begin{bmatrix} v_{x1}^0 & v_{x1}^1 & \cdots & v_{x1}^m \\ v_{x2}^0 & v_{x2}^1 & \cdots & v_{x2}^m \\ \vdots & \vdots & \ddots & \vdots \\ v_{xn}^1 & v_{xn}^1 & \cdots & v_{xn}^m \end{bmatrix} \begin{bmatrix} \omega_{i0} \\ \omega_{i1} \\ \vdots \\ \omega_{im} \end{bmatrix} \right\|_2, \tag{21}$$

where $\|\cdot\|_2$ represents Euclidean norm and

$$y_i = \begin{bmatrix} y_{i1} \\ y_{i2} \\ \vdots \\ y_{in} \end{bmatrix} \in \mathbb{R}^n, \tag{22}$$

$$V = \begin{bmatrix} v_{x1}^{\,0} & v_{x1}^{\,1} & \cdots & v_{x1}^{\,m} \\ v_{x2}^{\,0} & v_{x2}^{\,1} & \cdots & v_{x2}^{\,m} \\ \vdots & \vdots & \ddots & \vdots \\ v_{xn}^{\,1} & v_{xn}^{\,1} & \cdots & v_{xn}^{\,m} \end{bmatrix} \in \mathbb{R}^{n \times (m+1)}, \tag{23}$$

$$\boldsymbol{\omega}_i = \begin{bmatrix} \omega_{i\,0} \\ \omega_{i\,1} \\ \vdots \\ \omega_{i\,m} \end{bmatrix} \in \mathbb{R}^{m+1}. \tag{24}$$

Subsequently, each error is weighted by $\boldsymbol{\beta} = [\beta_1, \ldots, \beta_n]^{\mathrm{T}} \in \mathbb{R}^n$, where $\beta_i$ represents the evaluated performance value of the corresponding data point, $i$, with higher values indicating more favorable data. We define the following total performance-weighted error:

$$\| \boldsymbol{\beta} \otimes (\boldsymbol{y}_i - \boldsymbol{V}\boldsymbol{\omega}_i) \|_2, \tag{25}$$

where $\otimes$ denotes the Hadamard product, which takes two matrices of the same size and returns a matrix in which each element is the product of the original elements (see S1 Appendix).

In PWLS, the $\omega_{ij}$ coefficients are determined so that they minimize the total performance-weighted squared error, $E_{PWLS}$. Given that the error for each data point is weighted by its evaluated performance, $\beta_i$, the errors are more suppressed for high-performing data points and permissible for low-performing data points. $E_{PWLS}$ is the square of Eq (25),

$$E_{PWLS} = \| \boldsymbol{\beta} \otimes (\boldsymbol{y}_i - \boldsymbol{V}\boldsymbol{\omega}_i) \|_2^2, \tag{26}$$

$$= (\boldsymbol{\beta} \otimes (\boldsymbol{y}_i - \boldsymbol{V}\boldsymbol{\omega}_i))^{\mathrm{T}} (\boldsymbol{\beta} \otimes (\boldsymbol{y}_i - \boldsymbol{V}\boldsymbol{\omega}_i)). \tag{27}$$

The partial derivative of $E_{PWLS}$ with respect to $\boldsymbol{\omega}_i$ is given by

$$\frac{\partial E_{PWLS}}{\partial \boldsymbol{\omega}_i} = 2(\boldsymbol{\beta} \otimes (\boldsymbol{y}_i - \boldsymbol{V}\boldsymbol{\omega}_i))^{\mathrm{T}} \frac{(\boldsymbol{\beta} \otimes (\boldsymbol{y}_i - \boldsymbol{V}\boldsymbol{\omega}_i))}{\partial \boldsymbol{\omega}_i}, \tag{28}$$

$$= 2(\boldsymbol{\beta} \otimes (\boldsymbol{y}_i - \boldsymbol{V}\boldsymbol{\omega}_i))^{\mathrm{T}} (-\boldsymbol{B} \otimes \boldsymbol{V}), \tag{29}$$

where $\boldsymbol{B}$ is an $n \times (m + 1)$ matrix that expands $\boldsymbol{\beta}$ along the horizontal axis and is defined as follows:

$$B = \begin{bmatrix} \beta_1 & \beta_1 & \cdots & \beta_1 \\ \beta_2 & \beta_2 & \cdots & \beta_2 \\ \vdots & \vdots & \ddots & \vdots \\ \beta_n & \beta_n & \cdots & \beta_n \end{bmatrix} \in \mathbb{R}^{n \times (m+1)}. \tag{30}$$

$E_{PWLS}$ attains its minimum value when

$$\frac{\partial E_{PWLS}}{\partial \boldsymbol{\omega}_i} = 0. \tag{31}$$

Thus, the objective of PWLS is to solve Eq (31) with respect to $\boldsymbol{\omega}_i$. Combining Eqs (29) and (31) yields

$$2(\boldsymbol{\beta} \otimes (\boldsymbol{y}_i - \boldsymbol{V}\boldsymbol{\omega}_i))^{\mathrm{T}}(-\boldsymbol{B} \otimes \boldsymbol{V}) \quad = 0, \tag{32}$$

$$(\boldsymbol{\beta} \otimes (\boldsymbol{y}_i - \boldsymbol{V}\boldsymbol{\omega}_i))^{\mathrm{T}}(\boldsymbol{B} \otimes \boldsymbol{V}) \quad = 0. \tag{33}$$

Using the property, $(\boldsymbol{U}\,\boldsymbol{V})^T = \boldsymbol{V}^T\boldsymbol{U}^T$, Eq (33) can be rewritten as

$$(\boldsymbol{B} \otimes \boldsymbol{V})^{\mathrm{T}}(\boldsymbol{\beta} \otimes (\boldsymbol{y}_i - \boldsymbol{V}\boldsymbol{\omega}_i)) \quad = 0, \tag{34}$$

$$(\boldsymbol{B} \otimes \boldsymbol{V})^{\mathrm{T}}(\boldsymbol{\beta} \otimes \boldsymbol{y}_i) - (\boldsymbol{B} \otimes \boldsymbol{V})^{\mathrm{T}}(\boldsymbol{\beta} \otimes \boldsymbol{V}\boldsymbol{\omega}_i) \quad = 0, \tag{35}$$

$$(\boldsymbol{B} \otimes \boldsymbol{V})^{\mathrm{T}}(\boldsymbol{\beta} \otimes \boldsymbol{V}\boldsymbol{\omega}_i) \quad = (\boldsymbol{B} \otimes \boldsymbol{V})^{\mathrm{T}}(\boldsymbol{\beta} \otimes \boldsymbol{y}_i). \tag{36}$$

Here, $\boldsymbol{\beta} \otimes \boldsymbol{V}\,\boldsymbol{\omega}_i$ in Eq (36) can be reformulated (see S1 Appendix) as follows:

$$\boldsymbol{\beta} \otimes \boldsymbol{V}\boldsymbol{\omega}_i = (\boldsymbol{B} \otimes \boldsymbol{V})\boldsymbol{\omega}_i. \tag{37}$$

By substituting Eq (37) into Eq (36), we obtain

$$(\boldsymbol{B} \otimes \boldsymbol{V})^{\mathrm{T}}(\boldsymbol{B} \otimes \boldsymbol{V})\boldsymbol{\omega}_i = (\boldsymbol{B} \otimes \boldsymbol{V})^{\mathrm{T}}(\boldsymbol{\beta} \otimes \boldsymbol{y}_i). \tag{38}$$

Consequently, the polynomial coefficient, $\boldsymbol{\omega}_i$, in $P_i$ can be computed as follows:

$$\boldsymbol{\omega}_i = ((\boldsymbol{B} \otimes \boldsymbol{V})^{\mathrm{T}}(\boldsymbol{B} \otimes \boldsymbol{V}))^{-1}(\boldsymbol{B} \otimes \boldsymbol{V})^{\mathrm{T}}(\boldsymbol{\beta} \otimes \boldsymbol{y}_i). \tag{39}$$

## Design of weight parameters $\boldsymbol{\beta}$

$\beta_i \in \boldsymbol{\beta}$ is the performance evaluated in the PWLS to weigh corresponding data $i$. In this study, we evaluate the performance of each data point in terms of energy efficiency using the CoT. We design $\beta$ such that it has a higher value when the CoT is low. Because the energy consumed in human walking depends on speed [50], it is unreasonable to compare CoT values between data with widely different speeds. Thus, we calculate $\beta$ relative to the average CoT measured around the generated walking speed. Because the dataset, $\{(v_{x1}, \boldsymbol{Y}_1, \mathrm{CoT}_1), \ldots, (v_{xi}, \boldsymbol{Y}_i, \mathrm{CoT}_i), \ldots, (v_{xn}, \boldsymbol{Y}_n, \mathrm{CoT}_n)\}$, is sorted by speed, the average CoT around the $j$th data point, $\overline{\mathrm{CoT}}_j$, is calculated as follows:

$$\overline{\mathrm{CoT}}_j = \frac{\mathrm{CoT}_{j-M} + \mathrm{CoT}_{j-M+1} + \ldots + \mathrm{CoT}_{j+M}}{2M+1}. \tag{40}$$

In this study, $M$ was set to 125, giving $2M + 1 = 251$, which is approximately 0.5% of the control parameter sets collected in the dataset. Using the average CoT value around the $i$th

data point, $\overline{\text{CoT}}_i$, we defined the weight for data $i$, $\beta_i$, as follows:

$$\beta_j = A^{\left(\dfrac{\overline{\text{CoT}}_j - \text{CoT}_j}{\overline{\text{CoT}}_j}\right)}, \tag{41}$$

where $A \geq 1$ denotes a constant. $\beta_i > 1$ indicates that the data are evaluated as favorable data and $\beta_i < 1$ as unfavorable data. For example, when $\text{CoT}_i < \overline{\text{CoT}}_i$ (i.e., generated gait was more energy-efficient), $\left(1 - \frac{\text{CoT}_j}{\overline{\text{CoT}}_j}\right)$ is calculated above 0, and this yielded $\beta_i > 1$. A smaller CoTi value in comparison to the average CoT value, $\overline{\text{CoT}}i$, results in the exponential increase of $\beta_i$ based on the $A$ value. Therefore, the parameter $A$ determines the strength of the bias toward higher-performing data points, i.e., a larger $A$ value indicates a greater bias toward favorable data.

### Simple application example of PWLS

In this section, we explain how PWLS computes polynomials, based on a simple illustrative example. We assume that there is a dataset of a control parameter $y_i$, comprising a total of $18 \times 9$ data points, each evaluated through the CoT as depicted in Fig 5. The horizontal axis indicates the generated walking velocity and the vertical axis indicates the control parameter value. The black-colored data points correspond to a lower CoT value of 0.5 (more efficient), while the grey-colored data points correspond to a higher CoT value of 1.0 (less efficient). Fig 5 illustrates polynomials calculated through PWLS. It can be found that the polynomials pass close to the data points with highly evaluated data points with larger $A$, which determines the strength of the bias toward highly- evaluated data points (Eq (41)). Thus, by assigning a greater weight to higher-performing data points, walking energy efficiency can be reinforced via regression.

## Results

In the optimization, the upper timestep for each evaluation trial was set to $T = 5,000$ steps (25 s). The weight coefficients in the objective cost function, $f$, were set as $\alpha_E = 5000$, $\alpha_v = 5$ and $\alpha_t = 1.0$. We collected the dataset by running two programs in parallel with $v_x^{tar}$ incrementally increase/decrease in 0.1 m/s intervals. Within one thread, $v_x^{tar}$ was initially set to 1.3 m/s and increased to $v_{x\,max}^{tar} = 2.0$ m/s. Within the other thread, $v_x^{tar}$ was initially set to 1.2 m/s and decreased to $v_{x\,min}^{tar} = 0.4$ m/s. For each $v_x^{tar}$, $\lambda = 20$ points were sampled per generation, with a generation number of $G = 300$, which we confirmed that optimization was converged (See S1 Appendix). This resulted in 102,000 evaluation trials. The other hyperparameters for CMA-ES were set to $\sigma^0 = 0.1$ and $\mu = 7$. $\lambda$ and $\mu$ are recommended value [46]. $\sigma^0$ was set to be large, but not out of the solution space in the optimization process. Data collection required approximately three days on a Lenovo ThinkPad E470 20H2S04L00. Out of all trials, we added approximately 50% to the dataset. Fig 6 illustrates the generated walking speeds and the CoTs in a dataset, indicating a concave quadratic relationship similar to human walking [50]. Using PWLS, we obtained $P_i$ (Fig 4). We found that at least a polynomial degree of $m = 6$ is required to generate gaits. Therefore, to compute polynomials with minimal computational cost and to avoid overfitting, we chose a polynomial degree of $m = 6$.

### Generated gaits

To stabilize walking before setting $v^{tar}$, we controlled the musculoskeletal model with parameters that produced a stable gait at 1.25 m/s for a distance of 2.0 m from the initial position. We

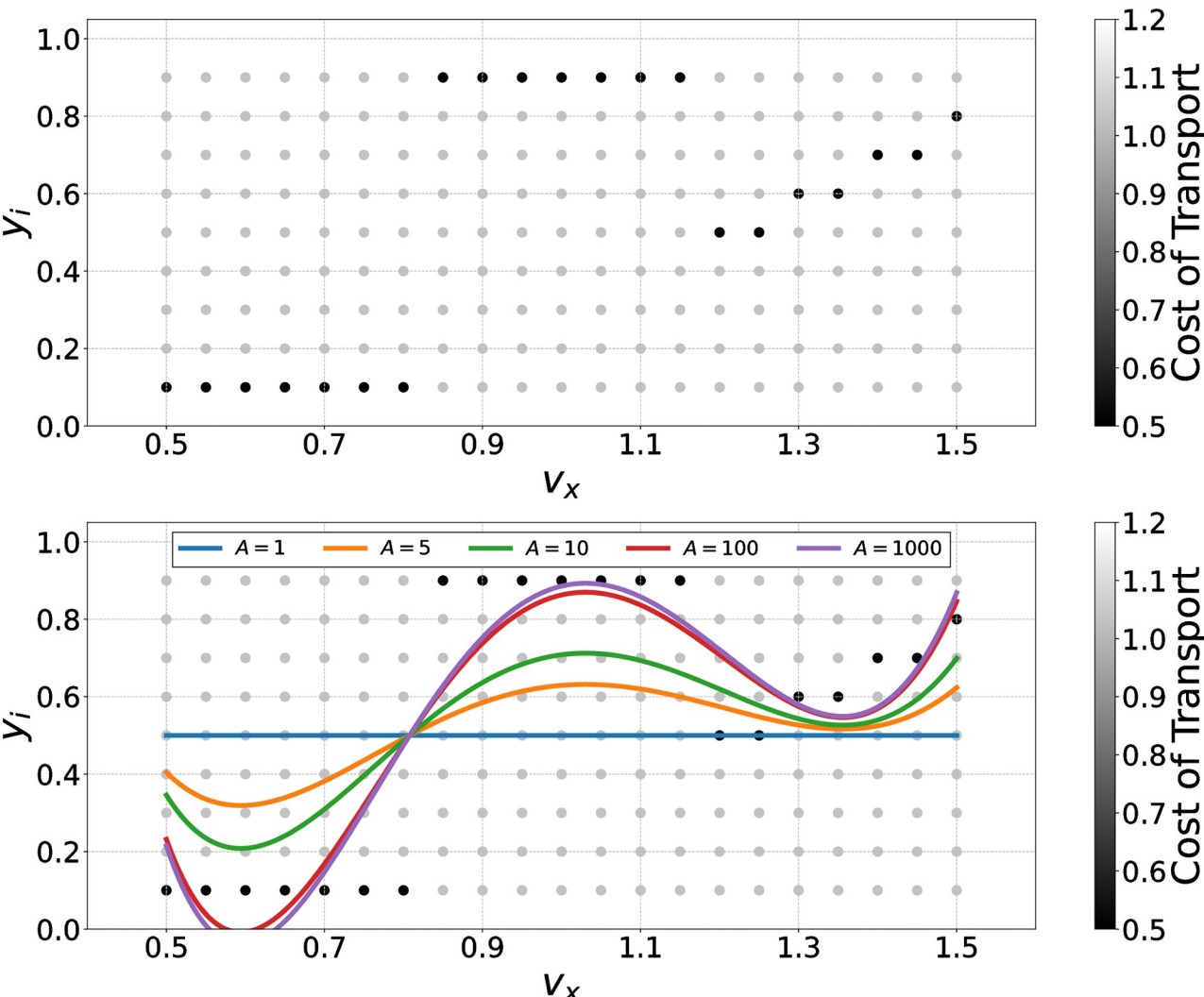

**Fig 5. The regression curve derived through PWLS with different *A* values.** Upper: The prepared dataset comprises $18 \times 9$ data points, each is evaluated through the CoT. The black-colored data points represent a lower CoT value of 0.5 (efficient), while the gray-colored data points represent a higher CoT value of 1.0 (inefficient). Lower: Calculated regression curves, with a polynomial degree set to 6.

found that steady walking was generated for $v_x^{tar} = 0.75 - 1.6$ m/s by using the optimized functions derived with $A = 1 - 10^6$ in Eq (41) (see S1 Video). Here, We defined steady walking as being able to walk more than 30 m without falling. The actual speeds were approximately to $v_x^{tar}$ (see S1 Appendix). Fig 7 displays snapshots of the generated gaits for $v_x^{tar}$ at 0.9 (slow), 1.25 (normal), and 1.6 (fast) m/s.

Fig 8 illustrates the ground reaction forces (GRF) and kinematics of the generated gait using the optimized functions derived with $A = 10^6$ and those of humans at corresponding speeds [51]. Although undershoots and overshoots were observed, the cross-correlation values (R) between the GRF of the generated and human gaits were consistently positive. In kinematics, the cross-correlation values for hip and knee joints were close to 1, whereas the value for the ankle joint was nearly 0.

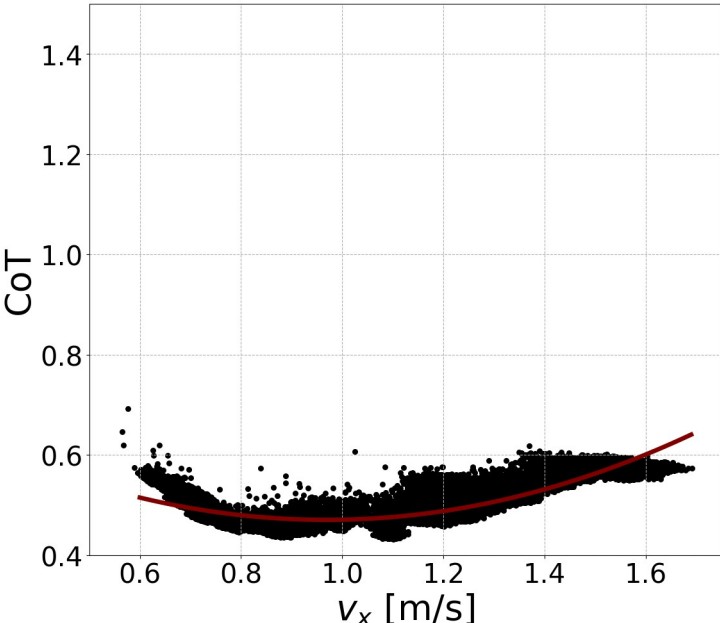

**Fig 6. Generated walking speeds and their CoTs in the dataset.** The CoT velocity value can be approximated by quadratic curves (red lines).

## Velocity control

In this section, we demonstrate the velocity control ability of the designed reflex-based controller. The upper graph in Fig 9 illustrates the velocity of the model with a change in $v_x^{tar}$ (dotted line) between the minimum and maximum values. The bottom graph in Fig 9 indicates an additional experimental result in which $v_x^{tar}$ was changed from 1.2 to 0.8 to 1.0 to 1.5, and finally to 1.3 m/s. In all cases, the rate of change of $v_x^{tar}$ was set to 0.05/s. As illustrated in the figures, the model regulates the velocities based on the target velocity.

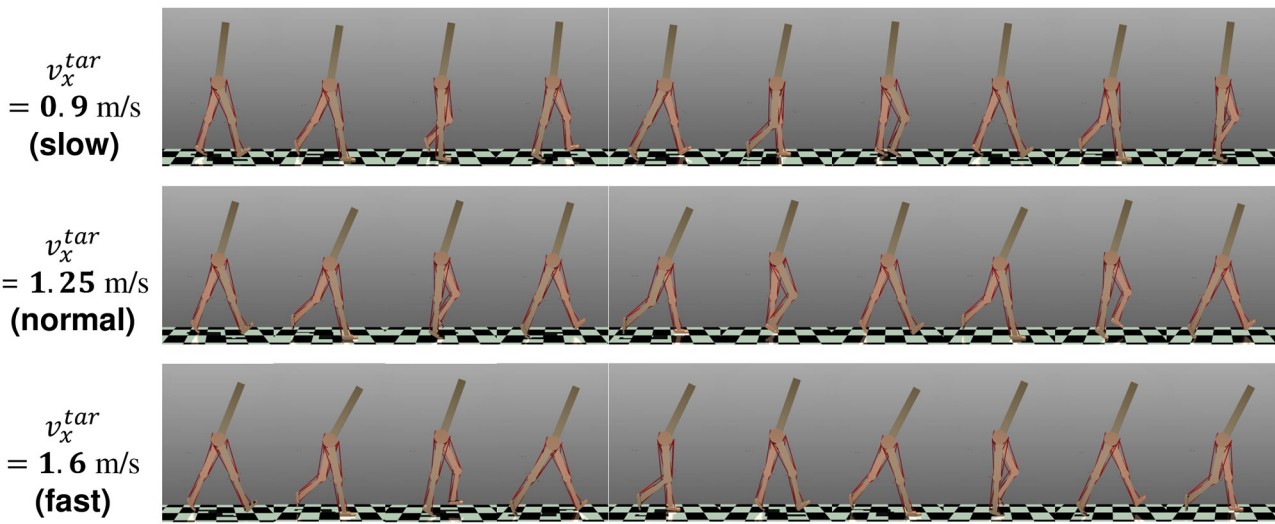

**Fig 7. Snapshots of the generated gait.** We captured data every 0.25 s for $v_x^{tar}$ at 0.9, 1.25, and 1.6 m/s. See S1 Video.

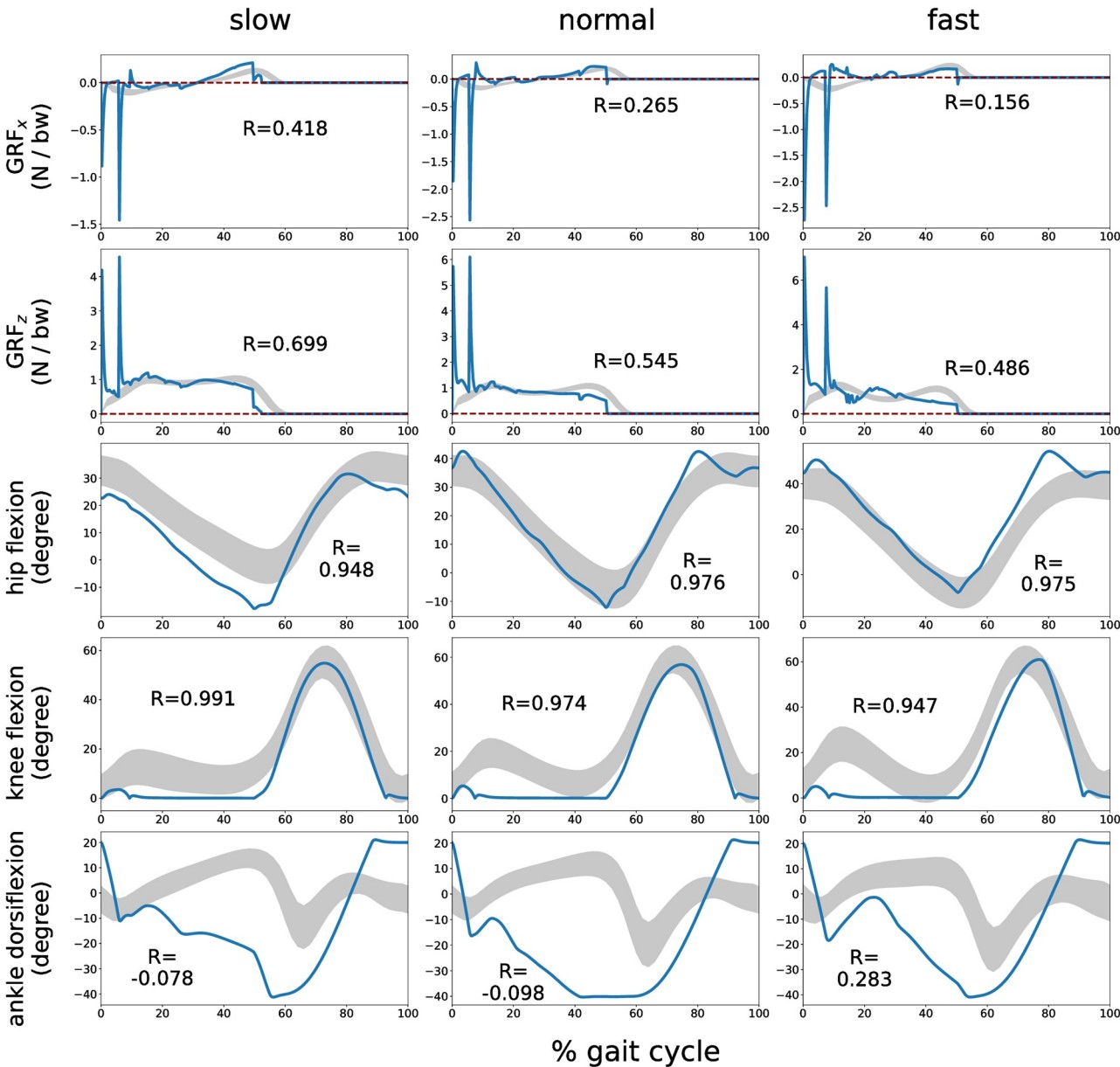

**Fig 8. GRFs and kinematics of the generated gait.** $v_x^{tar}$ was set to 0.9 (slow), 1.25 (normal), and 1.6 m/s (fast). Blue lines depict the generated gait, while gray lines represent those of humans (mean±1 s.d.) [51]. GRFs values are normalized by the body weight. $R$ denotes the cross-correlation value.

## Performance evaluation of PWLS

In this section, we evaluate the optimization performance of the proposed polynomial regression (PWLS) method. PWLS calculates regression curves by weighting higher-performing (i.e., energy-efficient) data points, in which $A$ determines the strength of the bias toward them. Fig 10 left shows the estimated CoT curves for the generated walking with different $A$ values ($A = 1$, 10, $10^2$, $10^4$, and $10^6$) in Eq (41). To quantify the performance of the PWLS, we defined

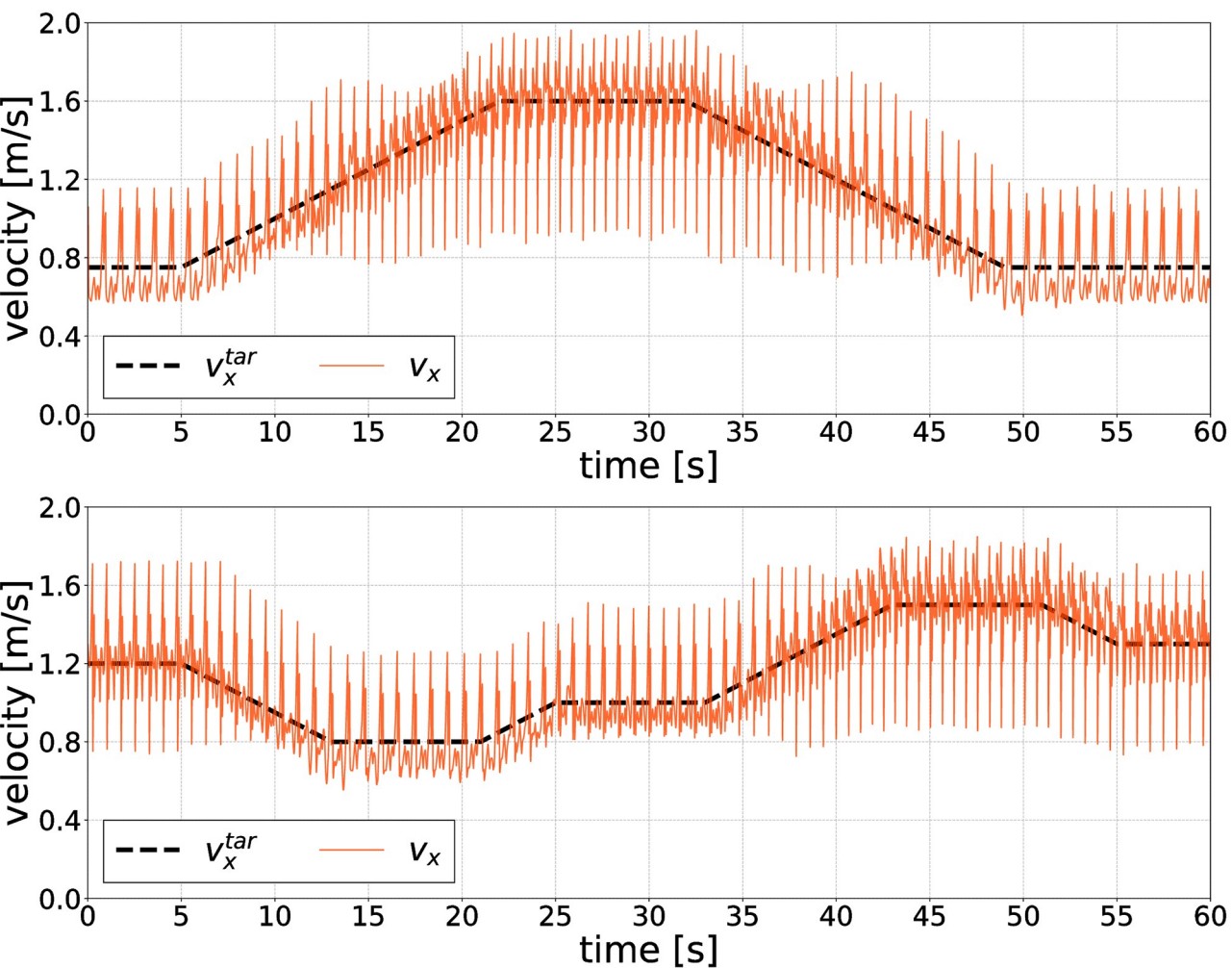

**Fig 9. The regression curve derived through PWLS with different $A$ values.** Upper: The prepared dataset comprises $18 \times 9$ data points, each is evaluated through the CoT. The black-colored data points represent a lower CoT value of 0.5 (efficient), while the gray-colored data points represent a higher CoT value of 1.0 (inefficient). Lower: Calculated regression curves, with a polynomial degree set to 6.

the integrated CoT value over the corresponding velocity range, $\int$CoT, as follows:

$$\int \mathrm{CoT} = \int_{v_{x\,min}}^{v_{x\,max}} \mathrm{CoT}\, dv_x, \tag{42}$$

where $v_{xmin}$ and $v_{xmax}$ represent the lower and upper limits of the velocity that the model can walk. For example, when the model can walk from $v_{xmin} = 0.7$ m/s to $v_{xmax} = 1.6$ m/s and the estimated CoT curve is represented $\mathrm{CoT} = 0.6v_x^2 - 1.2v_x - 0.1$, $\int$CoT is calculated as follows:

$$\int \mathrm{CoT} = \int_{v_{x\,min}=0.7\,\mathrm{m/s}}^{v_{x\,max}=1.6\,\mathrm{m/s}} \left(0.6v_x^2 - 1.2v_x - 0.1\right) dv_x \simeq 0.556. \tag{43}$$

Fig 10 right presents relative $\int$CoT value of the generated gaits with different $A$ values. Regardless of the $A$ value, $v_{xmin}$ (=0.6 m/s) and $v_{xmax}$ (=1.6 m/s) were set to be identical. We

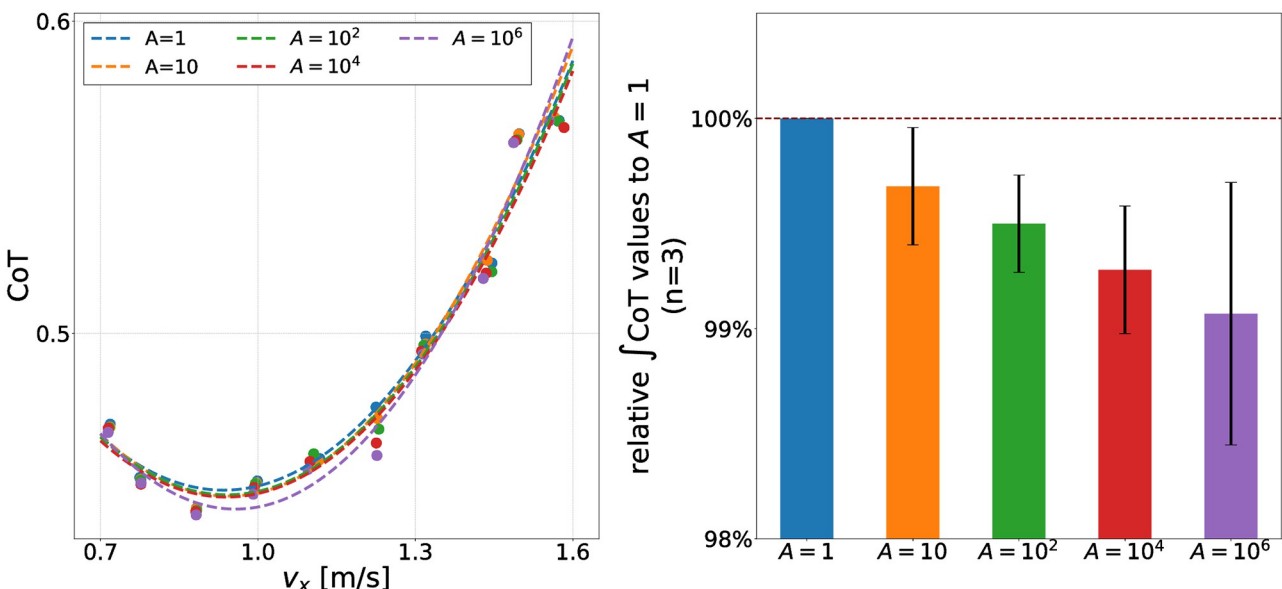

**Fig 10. Contribution of the PWLS to gait generation. Left: Estimated CoT curves for generated gaits with different $A$ values. Right: Relative $\int$CoT value of the generated gaits with different $A$ values (n = 3).** The generated gait with $A = 1$ is the baseline. $\int$CoT is the integral of the estimated CoT curves from the lower limit velocity (=0.7 m/s) to the upper limit velocity (=1.6 m/s).

found that lower $\int$CoT for larger $A$: more energy-efficient walking was generated with larger $A$. However, excessive $A$ resulted in increased instability: we could not find steady walking at specific $v_x^{tar}$ values for $A > 10^8$.

## Identifying factors essential for improving energy efficiency

In this study, we aim to identify the key reflex circuits that influence energy-efficient walking across a wide range of velocities. First, to identify the essential factors for improved energy efficiency, we substituted the optimized function of each control parameter that derived at $A = 10^6$, which achieved more energy-efficient walking over a wide range of velocities, in the gait with $A = 1$. Fig 11 illustrates this approach. Fig 12 illustrates the absolute change in

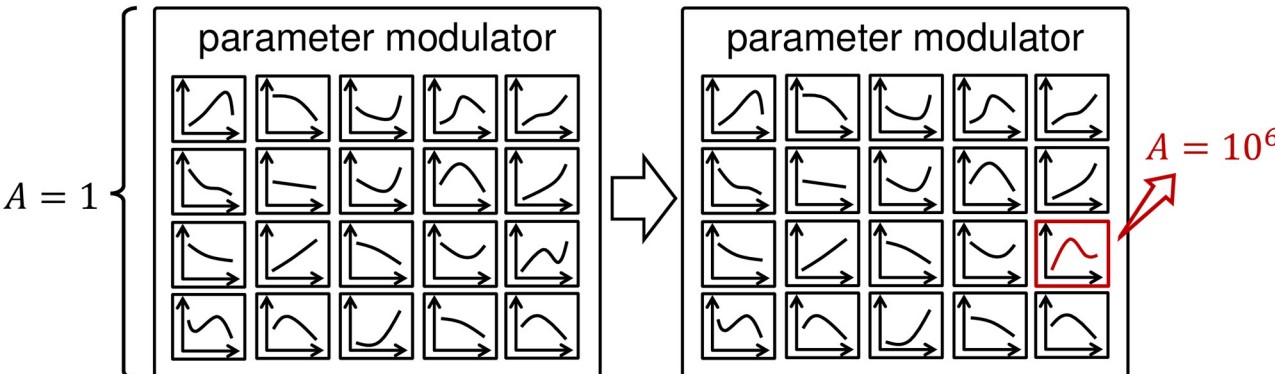

**Fig 11. Illustration of the replacement of the optimized function. Left: Before the replacement, all functions were derived with $A = 1$. Right: Only the target control parameter function (red) was replaced with $A = 10^6$. Functions with other parameters were not replaced.**

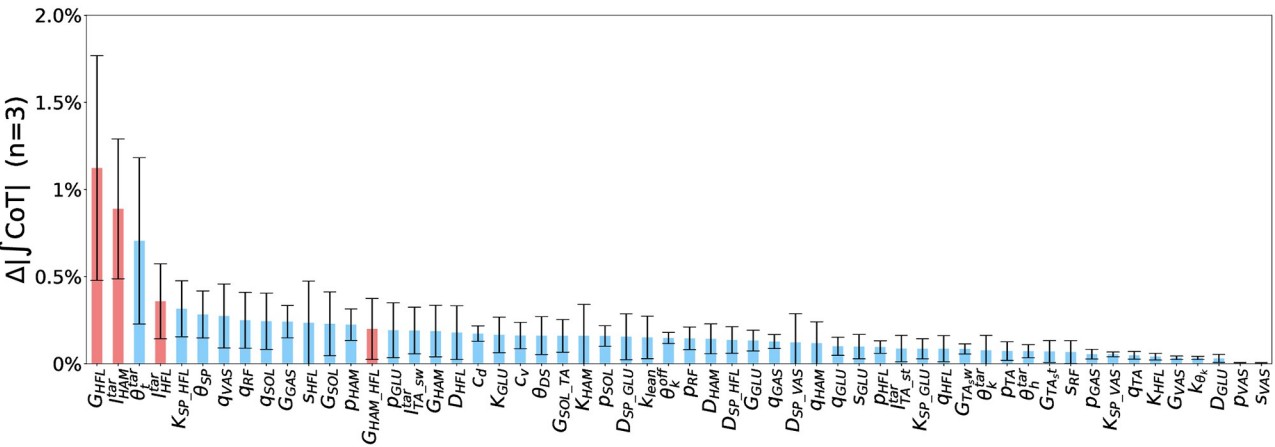

**Fig 12. Absolute change in $\int$CoT values when the optimized function of each parameter was substituted to that derived with $A = 10^6$ in the generated gait with $A = 1$ ($n = 3$).** The red bars represent control parameters associated with the reflex circuits under focus.

percentage of the $\int$CoT values, $\Delta|\int$CoT$|$, to the generated gait with $A = 1$. From the results, we found that $G_{HFL}$, $l_{HAM}^{tar}$ and $\theta_t^{tar}$ are the three parameters that changed $\Delta|\int$CoT$|$ value by more than 0.5%. This result suggests that reflex circuits that involve these parameters may have a significant impact on energy-efficient walking across a wide range of velocities. The two length-feedback reflex circuits, $G_{HFL}(l_{HFL} - l_{HFL}^{tar})$ and $G_{HAM\_HFL}(l_{HAM} - l_{HAM\_HFL}^{tar})$, is the primary circuits involved in these parameters, $G_{HFL}$ and $l_{HAM}^{tar}$: both reflex circuits stimulate HFL muscle during the swing phase (Fig 13). $\theta_t^{tar}$, which is the reference angle of the torso, changed $\Delta|\int$CoT$|$ value

**circuit 1: $G_{HFL}(l_{HFL} - l_{HFL}^{tar})$**

stimulate HFL, **facilitate** swing

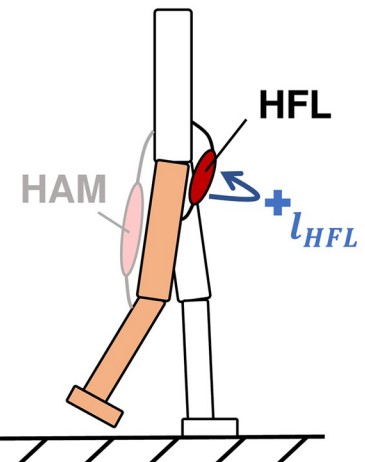

**circuit 2: $G_{HFL\_HAM}(l_{HAM} - l_{HFL\_HAM}^{tar})$**

stimulate HFL, **inhibit** swing

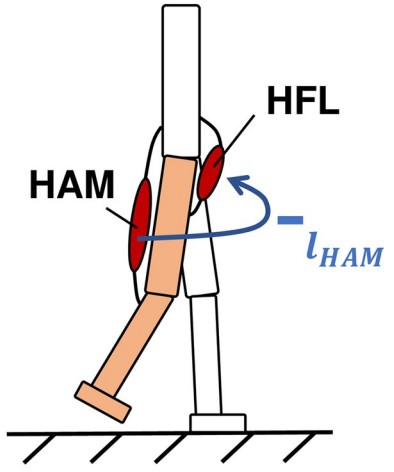

**Fig 13. Key reflex circuits for energy-efficient gait.** $G_{HFL}(l_{HFL} - l_{HFL}^{tar})$ is the positive length feedback stimulating HFL to swing the leg forward. $G_{HAM_HFL}(l_{HAM} - l_{HAM_HFL}^{tar})$ is the negative length feedback inhibiting the HFL proportional to the stretch of the HAM in the swing phase. In these terms, $G$ represents the gain, $l$ represents the length of the muscle, and $l^{tar}$ represents the constant target length. Both reflex circuits are active only in the swing phase.

of the third. This parameter was not related to a specific reflex circuit but was shared with several reflex circuits.

## Key reflex circuits for energy-efficient walking over wide range velocity

The aforementioned results suggest that $G_{HFL}(l_{HFL} - l_{HFL}^{tar})$ and $G_{HAM\_HFL}(l_{HAM} - l_{HAM\_HFL}^{tar})$ could be the parts of the essential reflex circuits on the wide-range velocity walking with maintaining energy efficiency. Therefore, we proceeded to investigate how modulating these reflex circuits affects the energy efficiency of the gait. We measured the change in $\int$CoT values when the optimized functions of the control parameters associated with $G_{HFL}(l_{HFL} - l_{HFL}^{tar})$ (reflex circuit 1) and $G_{HAM\_HFL}(l_{HAM} - l_{HAM\_HFL}^{tar})$ (reflex circuit 2) were replaced with those derived with $A = 10^6$ in the generated gait with $A = 1$ (a baseline), similar to Fig 11, and the result is shown in Fig 14. The cyan and olive bar represents the relative $\int$CoT value when we replaced only the optimized functions associated with reflex circuits 1 and 2, respectively. The pink bar represents the relative $\int$CoT value when we replaced the optimized function of both reflex circuits. The results indicated that the modulation of only these two reflex circuits resulted in a comparable or even more substantial reduction in $\int$CoT value compared to the modulation of all 56

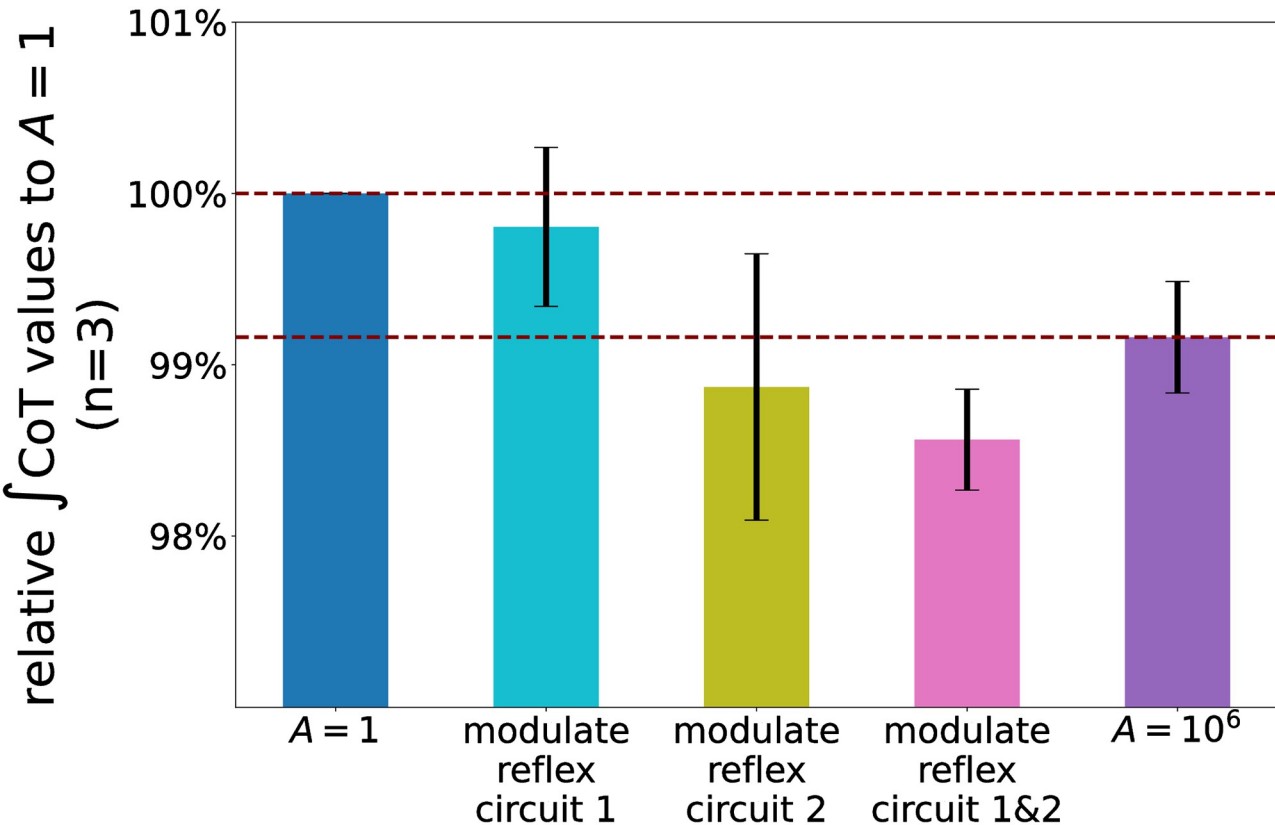

**Fig 14. Relative $\int$CoT values to the generated walking with $A = 1$ when modulating identified reflex circuits (n = 3).** The cyan and olive bar represents the relative $\int$CoT value when only the optimized function of control parameters associated with reflex circuit 1 (i.e. $G_{HFL}$ and $l_{HFL}^{tar}$) and reflex circuit 2 (i.e. $G_{HAM\_HFL}$ and $l_{HAM_HFL}^{tar}$) were replaced with those derived with $A = 10^6$ in the generated gait with $A = 1$, respectively. The pink bar represents when the optimized function of the control parameters associated with both reflex circuits (i.e., $G_{HFL}$, $l_{HFL}^{tar}$, $G_{HAM\_HFL}$, $l_{HAM_HFL}^{tar}$) were replaced. The purple bar indicates the value in the generated gait using the optimized functions derived with $A = 10^6$. When circuit 2 was modulated, some generated gaits did not follow the specific input $v_{vel}^{tar}$, with their velocities being changed by more than 0.05 m/s. Data acquired from the target velocity that resulted in these outliers were excluded from the calculation of estimated CoT curves.

parameters for the case of $A = 10^6$ shown in the purple bar. This finding strongly suggests that the two length-feedback reflex circuits, $G_{HFL}(l_{HFL} - l_{HFL}^{tar})$ and $G_{HAM\_HFL}(l_{HAM} - l_{HAM\_HFL}^{tar})$, are the key reflex circuits of the energy efficiency in the reflex-based bipedal walking control.

To elucidate the mechanism behind the improved energy efficiency, we examined the change in energy consumption by individual muscles, muscle activation patterns, and stimulation signal for the corresponding muscle when we modulated the two identified reflex circuits. Fig 15 shows the energy consumed by individual muscles during a 30 m walk at $v_x^{tar} = 1.0$. We found that energy consumption of the HFL, GLU, and HAM muscles was significantly reduced after reflex circuits 1 and 2 were modulated. Fig 16 shows the muscle activations of the bipedal model for $v_x^{tar} = 1.0$ m/s. HFL muscle was mainly activated in the swing phase while GLU and HAM muscles were activated in the stance phase. Fig 17 illustrates the stimulation signal applied to HFL muscle during the swing phase before (blue line) and after the modulation (pink line) of reflex circuits 1 and 2: the stimulation signal applied to HFL muscle decreased after the modulation of these identified reflex circuits.

## Robustness to parameter changes

Under various parameter conditions, leg length, sensory time delay, and weight coefficients for the cost function, we have validated the reliability and robustness of the identified two reflex circuits concerning their impact on energy-efficient walking over a wide range of velocities:

- shorten segment length by 20% (different body structure)

- double the time delay of sensory information transmission to the controller (different neural system)

- change the weight coefficients in the objective cost function (Eq (31)), $\alpha_E = 2500$, $\alpha_v = 5$, and $\alpha_t = 0$ (different cost function)

Fig 18 illustrates the relative $\int$CoT values to the baseline ($A = 1$) under various conditions. We found that modulating reflex circuits 1 and 2 resulted in decreased $\int$CoT values, comparable to the result for all 56 parameter modulation (generated gaits can be seen in S1 Video).

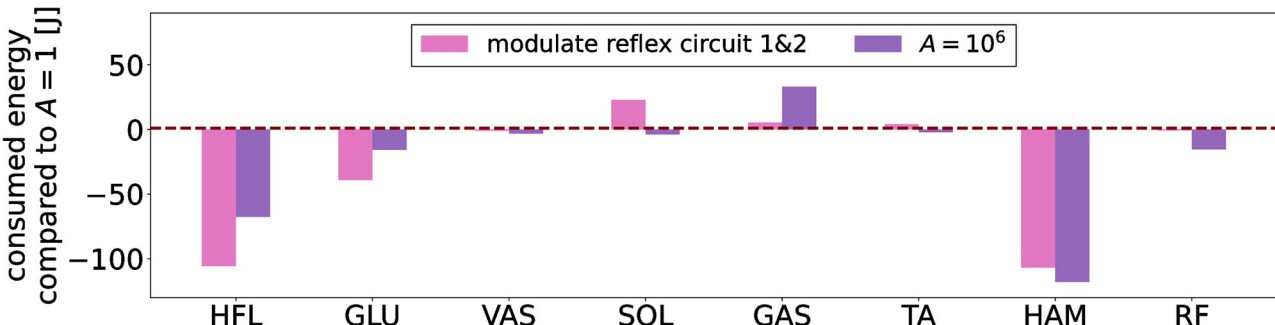

**Fig 15. Energy consumed by individual muscles compared to the generated gait with $A = 1$ during a 30 m walk.** $v_x^{tar}$ was set to 1.0 m/s. The pink bars represent the values when the optimized functions of the control parameters associated with reflex circuits 1 and 2 are replaced with those derived with $A = 10^6$ in the gait generated with $A = 1$, and the purple bars represent the values in the gait with $A = 10^6$. The negative value indicates that the energy consumption at the muscle was reduced compared to the generated gait with $A = 1$, while the positive value indicates increased energy consumption.

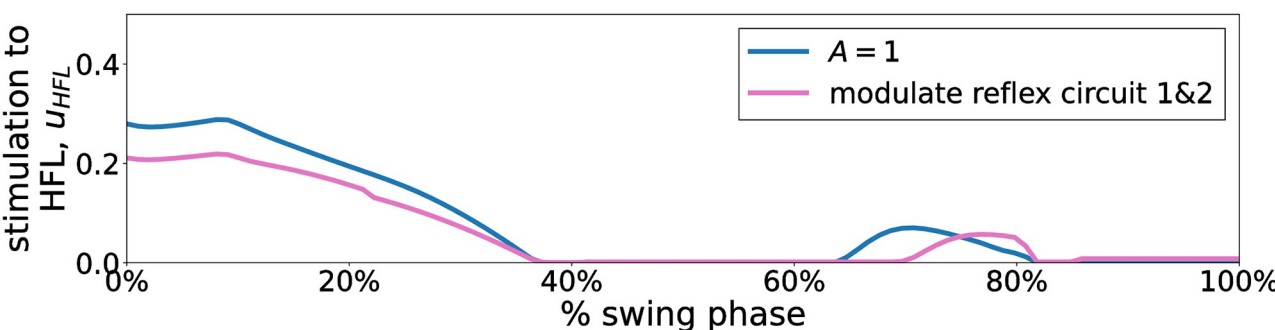

**Fig 16. Individual muscle activations.** The gait was generated using the optimized functions derived with $A = 10^6$ and $v_x^{tar}$ was set to 1.0 m/s. The red bar indicates the timing of the transition from stance to swing.

**Fig 17. Stimulation to the HFL in the swing phase before and after modulating reflex circuit 1&2.** $v_x^{tar}$ was set to 1.0 m/s. The blue and pink lines are the stimulation to the HFL, which does not drop below 0, at generated gait with $A = 1$ and after the optimized function of the control parameters associated with reflex circuit 1 and 2 are replaced with those derived with $A = 10^6$, respectively.

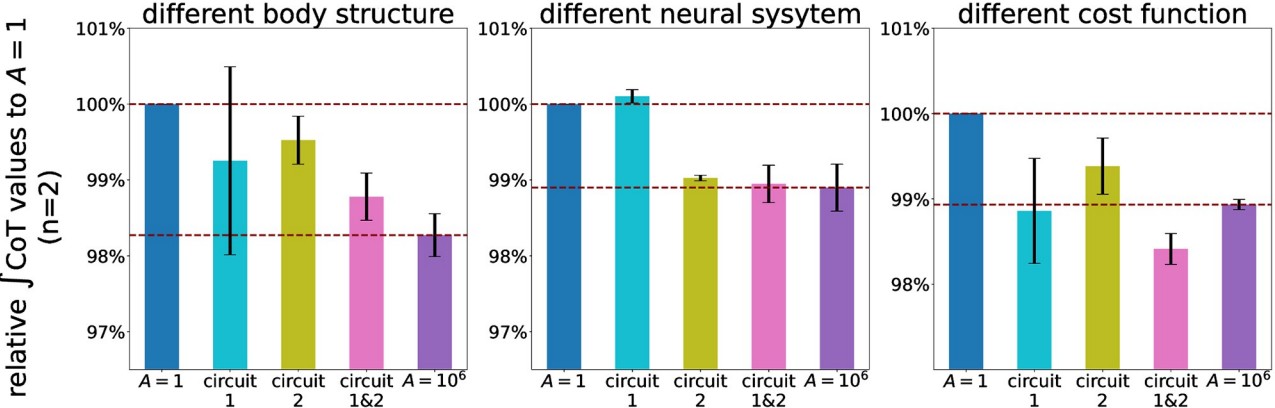

**Fig 18. Relative $\int$CoT values to the generated walking with $A = 1$ under different setting parameters (n = 2).** In a different body structure setting, the segment lengths of the model were shortened by 20%. In a different neural system setting, the time delay of the sensory information to the controller was doubled. In this setting, stable gaits were not generated for $A > 100$. In a different cost function setting, the weight coefficients in the objective function (Eq (31)) were changed to $\alpha_E = 2500$, $\alpha_v = 5$, and $\alpha_t = 0$.

## Discussion

This study aimed to extend the reflex-based control system including velocity control and identify key reflex circuits that play a significant role in energy-efficient walking. We demonstrated that a musculoskeletal model driven by reflex-based control can achieve controlled speeds based on the input target velocity. Subsequently, by utilizing the generated gaits of varying energy efficiencies, we identified two key reflex circuits having a significant impact on CoT values.

We found that the proposed PWLS regression method optimizes the parameter modulator for the reflex-based controller to reproduce more energy-efficient walking across a wide range of velocities. We verified that the $\int$CoT values were decreased by giving the bias to the high-performing data in calculating regression curves as shown in Fig 10. These results demonstrate that the PWLS fitting method is more adaptive in evaluating each data point as a weight for performance. However, we also found that the walking generated at an excessively large $A$ value resulted in less stability, resulting in a fall at specific target velocities. This can be attributed to insufficiently fast leg swing. Putting the swing leg fast enough in front of the stance leg is essential to prevent falling down [52]. As illustrated in Fig 17, the stimulation applied to the HFL during the swing phase decreased with larger $A$. Consequently, this led to a slow hip flexion resulting in less stability.

As shown in Fig 14, we identified that $G_{HFL}(l_{HFL} - l_{HFL}^{tar})$ and $G_{HAM\_HFL}(l_{HAM} - l_{HAM\_HFL}^{tar})$ significantly contributed to improve the energy efficiency of the gait generated through reflex-based control. Furthermore, this conclusion has been robustly validated across various settings, including a different body structure, different neural systems, and different cost functions, as shown in Fig 18. Our findings elucidated that the modulation of these reflex circuits resulted in a reduction of stimulation to the HFL during the swing phase, subsequently leading to reduced energy consumption by the HFL, GLU, and HAM (Fig 15). HFL expended less energy due to reduced stimulation. GLU and HAM were mainly activated during the stance phase (Fig 16), and these two muscles are used to maintain the torso balance in the stance phase [28]. Reduced HFL activity during the swing phase leads to less energy consumption at the contralateral GLU and HAM because the work to compensate for torso acceleration caused by HFL is decreased. Therefore, it can be concluded that minimizing the stimulation to the

HFL during the swing phase while ensuring sufficient hip swing to prevent falls is essential to improve energy efficiency in reflex-based walking control. This not only reduces HFL activity but also the effort of the GLU and HAM, which compensates for torso acceleration caused by HFL.

The generated walking has several characteristics that demonstrate its biomechanical validity in comparison to the human gait. (i) the $R$ values of the hip and knee kinematics were close to 1 ($R > 0.94$ in all cases); (ii) The trajectory of the hip segment height from the ground exhibited a sinusoidal pattern (see Fig. S7 in S1 Appendix) [53]; (iii) we observed both heel strike and toe-off events [54] (see attached movie file); and (iv) a quadratic relationship between walking velocities and CoT [50] (Fig 10). These features are similar to those of human walking from a biomechanical perspective, proving that our simulations do not compromise the dynamically reasonable properties of human walking. Although we agree that reducing the number of control parameters would be essential and helpful for understanding the mechanism underlying walking, we used Wang's extended model rather than Geyer's original model to robustly generate walking without losing biomechanical explanatory ability.

The results of this paper have some limitations concerning similarity to human walking. First, overshoots and undershoots were observed in the measured GRFs (Fig 8) and time evolution of the walking velocities (Fig 9). In the GRF profiles, we found two undershoots in $GRF_x$ and two overshoots in $GRF_z$. The first occurred at heel strike, and the second occurred at toe-off of the contralateral leg. These overshoots and undershoots can be suppressed by using a lower-impedance ground. However, employing a lower-impedance ground resulted in foot penetration into the ground (refer to S1 Appendix). While the extreme GRF peaks are not biomechanically meaningful, we strongly believe that it is difficult to accurately model contact in a simulation environment [55, 56]. Moreover, we did not observe abrupt or significant changes in joint kinematics attributable to these extreme GRF peaks. Thus, given the difficulty of modeling contact in the simulation environment, the extreme peaks in GRF do not significantly affect kinematics. We conclude that the effects of the extreme peaks of GRF on walking in the musculoskeletal model do not negatively affect the biomechanical meaning of the findings obtained in this study. Second, the cross-correlation values of ankle dorsiflexion were close to 0 compared to hip and knee joints, as shown in Fig 8. At knee joints, the model straightened the knee earlier in the stance phase compared to humans. This strategy is known to generate more efficient solutions in gait optimization [32]. The previous study also showed low cross-correlation values at ankle joints when optimized to minimize energetic cost [32]. Third, we designed the cost function employed in this study to have minimal task terms to generate a human-like gait with more weight on the energy efficiency-related term. This approach optimized the control parameters to improve energy efficiency, consequently allowing the identification of factors that are significant to the energy efficiency of the generated gait over a wide range of walking velocities. More complex cost functions [29, 34, 36] may be required to generate human-like ankle joint kinematics. Finally, although this study successfully demonstrated the implementation of velocity control in a reflex-based control system, it took 20 s to transit walking speed by 1.0 m/s without falling. On the other hand, humans can adjust their walking speed by 1.0 m/s in less than 2 s [57]. Consequently, our control framework lacks the component to change walking speeds rapidly while ensuring stability. Extending the proposed framework is one of the future works to achieve velocity control as fast as humans.

Human locomotion involves complex interactions between descending supraspinal commands, interconnected spinal circuits involving reflexes and CPGs, and the musculoskeletal system. This study primarily focuses on reflexes in the spinal cord. Therefore, we must add other control components for fast walking velocity control. Moreover, we constrained the

bipedal model motion into the sagittal plane. Hence, the extension of the motion into three dimensions is another essential future work to apply the controller to engineering applications.

## Conclusion

Reflex mechanisms contribute significantly to the generation of stable and energy-efficient walking. However, a major limitation of generating gaits in musculoskeletal models through reflex-based control is the difficulty in precisely regulating velocity due to the large number of control parameters that need to be properly tuned. Extending reflex-based systems to affect velocity controls is essential to explore the reflex modulation mechanism and to understand its energy-efficient maintenance mechanism across a wide range of velocities. Furthermore, the development of energy-efficient control over a wide range of velocities in the reflex-based system will facilitate advanced engineering applications. Therefore, we developed a reflex-based control framework that enables the regulation of walking velocity over a wide range of velocities. Our parameter modulation method using PWLS that calculates the control parameters in response to a target velocity while optimizing efficiency successfully demonstrates generating walking gaits from 0.7 to 1.6 m/s. Furthermore, after a detailed analysis of the parameter modulator in a reflex-based system, we identified that the modulations of two reflex circuits, $G_{HFL}(l_{HFL} - l^{tar}_{HFL})$ and $G_{HAM\_HFL}(l_{HAM} - l^{tar}_{HAM\_HFL})$, improve energy efficiency of the gait. The coordinated activity in the swing phase between the HFL and HAM reduced the stimulation applied to HFL during the swing phase, which not only caused the reduction of HFL activity but also alleviated the effort of GLU and HAM that compensates for the torso acceleration induced by the HFL. This research will inspire future investigations into reflex mechanisms and facilitate the development of advanced walking control systems for practical applications, such as gait-assisted exoskeletons and prosthetic legs, and robot control.

## Supporting information

**S1 Appendix. Detailed description of the main text.**
(ZIP)

**S1 Video. Video of the generated gait.**
(MP4)

## Author Contributions

**Conceptualization:** Shunsuke Koseki.

**Data curation:** Shunsuke Koseki.

**Formal analysis:** Shunsuke Koseki.

**Funding acquisition:** Mitsuhiro Hayashibe, Dai Owaki.

**Investigation:** Shunsuke Koseki.

**Methodology:** Shunsuke Koseki, Mitsuhiro Hayashibe, Dai Owaki.

**Project administration:** Shunsuke Koseki, Dai Owaki.

**Resources:** Shunsuke Koseki.

**Software:** Shunsuke Koseki.

**Supervision:** Mitsuhiro Hayashibe, Dai Owaki.

**Validation:** Shunsuke Koseki.

**Visualization:** Shunsuke Koseki.

**Writing – original draft:** Shunsuke Koseki.

**Writing – review & editing:** Shunsuke Koseki, Mitsuhiro Hayashibe, Dai Owaki.

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
