## [Decision Letter · Decision Letter 0]

16 Jun 2023

Dear Mr Koseki,

Thank you very much for submitting your manuscript "Identifying essential factors for energy-efficient walking control across a wide range of velocities in reflex-based bipedal systems" for consideration at PLOS Computational Biology. As with all papers reviewed by the journal, your manuscript was reviewed by members of the editorial board and by several independent reviewers. In light of the reviews (below this email), we would like to invite the resubmission of a significantly-revised version that takes into account the reviewers' comments.

Specifically, we have the general consensus that questions 1) the robustness of the essential conclusions considering the many control parameters this work introduced, 2) even if holds up, how to generalise this work to a more biologically relevant context (e.g., neural control of human locomotion), and 3) the novelty of the work considering what others have done. While appreciating your efforts and acknowledging the importance of the problem that you and your colleagues tried to address, we feel that the paper, as it currently stands, lacks a convincing theoretical framework that is necessary to be considered for publication in PLoS Computational Biology. We sincerely hope that the detailed technical reviews below could be of help for you to improve the paper. 

We cannot make any decision about publication until we have seen the revised manuscript and your response to the reviewers' comments. Your revised manuscript is also likely to be sent to reviewers for further evaluation.

Sincerely,

Jian Liu

Academic Editor

PLOS Computational Biology

James O'Dwyer

Section Editor

PLOS Computational Biology

Reviewer's Responses to Questions

**Comments to the Authors:**

Reviewer #1: The authors present an optimization approach which generates polynomial based functions for numerous control parameters in dependence of desired walking speed for a neuro-muscular model of human walking.

While acknowledging the importance of neuro-muscular feedback the authors make numerous changes which deviate substantially from a reflexive controller without clearly motivating them. CPGs, which are prominently featured in the introduction do not seem to play a role in the presented approach. The introduction of desired positions, constant control parameters and PD-control for selected muscles highly increases the number of control parameters. The motivation and influence these changes have and their individual impact is not explained.

The authors have implemented their model in Mujoco but it is not clear to the reviewer, if the simulation generates physically feasible results. At least, examples of ground reaction forces should be shown to demonstrate a sound simulation.

While the question is interesting, and has been to an extend already answered by Dzeladini, and the optimization approach is interesting, the reviewer does not understand the reason for not applying this optimization to the original, reflex based model and can only assume that the required adaptions are necessary to make the simulation work. This however would largely reduce the optimization's impact on general problems.

According to their discussion, the authors fail to clearly motivate their extensive extension of the reflex-based control system, and also did not clearly identify relevant reflex-control parameters, as many parameters seem to have no influence and of those who have, many have been introduced by the authors without a clear functional explanation. The term reflex rules seem to be very losely used as any type of feedback. Especially the direct inclusion of joint angles into the feedback rule deviates from established proprioceptive understanding of muscle-tendon units and their control.

In general, the paper's organisation makes it hard for the reader to follow the authors reasoning and clearly determine what has been taken from previous work and what is the authors contribution. Some contradictions, e.g. the supposedly very small values for p and q which at times can still reach magnitudes of 0.1 and more and are not compared to the other terms of the controllers, make the comprehension even harder.

In order to improve the paper and present the core idea, the authors should reorganize the paper and clearly motivate the changes they make to a model, which should be proven to be dynamically sound.

Reviewer #2: 

This paper introduces a new optimization approach that generates reflex-based 2D bipedal walking simulations capable of regulating the walking velocity across a wide range (0.5-1.5m/s) while minimizing the cost of transport (CoT). While the authors present their rather sophisticated modeling, simulation, and optimization processes in a clear and descriptive manner, I have some major concerns about this manuscript in that 1) the investigation is largely exploratory, and 2) the findings do not generalize to broader (e.g., more realistic) context, with less implications to the biological (neural) control of human locomotion. I elaborate on these points in detail in the comments below, along with some suggestions.

<major comments="">

1. The investigation remains largely exploratory and lacks any clear technical objective to be validated or a scientific question to be answered. The study exploits large (i.e., high-dimensional) solution space with rather arbitrary parameter choices to produce 2D walking kinematics. In contrary to what the authors identify as a problem (L1-2; L33-36), they still employ a reflex-based controller that involves “vast number of control parameters” (54 in total) and quite a substantial computational overhead (L236-242; L242-243), but add another layer of optimization, namely the performance-weighted least squares (PWLS) parameter modulation method to optimize these control parameters as a function of target velocity while minimizing the CoT. In doing so, many parameters are chosen based on “trial and error (L230)” or “empirical selection (L246-247)”; some even discussed as main findings (L252-252; L309-319) while many reported without any justification or rationale (L236-242). Such unprincipled, unsystematic exploration led to only illustrative presentation of the search/optimization results (L235-254; L263-280; Fig.4-7), without any criteria (or expectation) by which the contribution of each step or choice can be assessed. On a related note, many questionable aspects of the results are left with no interpretation or discussion including the outlier in the solution space (Fig.4), under-/overshoot or errors in terms of tracking desired velocity (Fig. 6), and seemingly subtle to no influence of the key parameter (the strength bias parameter A; Fig. 7, especially on the final optimized polynomial functions for each control parameter Fig. 7C). Lastly, but not the least - in fact most importantly - when identifying the essential factors for improving energy efficiency, which was presented as one of the main goals for this work (in the Title, Abstract, Summary, Introduction, Discussion, and Conclusion), the authors simply when about replacing the result in optimized parameter modulator from one condition to another (A=1, A=800) and evaluated its impact on CoT, which only resulted in <3% changes in few control parameters (or 5% at most, in one case). I am completely not convinced whether this is the most proper (systematic and rigorous) way to test and demonstrate the key point of the entire work!

2. The findings are unlikely to generalize to broader (e.g., more realistic) context and the implications do not shed much light on the biological (neural) control of human locomotion. This work employs a 14-segment, 9-DoF (degrees of freedom) model in the sagittal plane. Furthermore, the control is purely based on spinal reflex circuitry from a set of few of representative muscles. This is not to say that simple models are wrong. Such simplified model has been used to elucidate important principles of human locomotion, as in many papers cited by the authors (e.g., Geyer & Herr 2010, IEEE TNSRE) and more (Dzeladini et al., 2014, Front Human Neurosci; Kuo 2002, Motor Control; Taga et al., 1991, Biol Cybern). However, the scope of this paper, including the justification for methodological choices and discussion of the results and implications, lacks any insights beyond the technical increment at hand, i.e., implementation of velocity regulation with extra layer of optimization. For example, one could ask what the implications are for assuming a control that solely relies on feedback (vs. passive mechanics, active feedforward, or their combination) or a fixed set of parameters tuned as a function of walking velocity while minimizing energetic cost (vs. other functional criteria such as stability or effort). On the other hand, what are the principles derived from the examined model that can still hold true in a more generalized context, such as under gravity, physical interaction with the environment (e.g., ground), 3D or whole-body dynamics, viscoelastic tendons, system dynamics (e.g., stability and robustness), in the presence of noise/variability or external disturbances, or across wider range of locomotive behaviors including healthy and pathological? The control of human locomotion involves complex interactions between descending supraspinal neural commands (e.g., corticospinal, reticulospinal), interconnected spinal circuits and its modulation on top of simple length- and force-dependent reflexes, as well as the musculoskeletal system and the environment (Rossignol et al., Physiol Rev, 2006). None of the above aspects are discussed than being listed as yet some other factors influencing human locomotion, only in the Introduction. Moreover, owing to the exploratory approach critiqued above, and due to the very nature of ‘optimal’ solutions (Sternad, 2018; Curr Opin Behav Sci) especially in such a high-dimensional parameter space, the findings even within the scope of this paper is unlikely to be generalizable other test conditions. While I agree that the current work holds potential for engineering applications such as control of robots and prosthetics (L10-12; L16-19; L372-373), I cannot agree that the current findings “shed light on the intricate mechanisms underlying human walking (Abstract)”.

In conclusion, the presented work in its current form seems to remain much at near the end of a developmental stage. In other words, I acknowledge that the first two goals/contributions (among three) that the authors claim (L62-65), i.e., successful implementation of the proposed method have been achieved, where the proposed approach demonstrates promising initial results. Nevertheless, I believe that the work is not yet qualified a significant contribution, where the approach is yet to be tested for its robustness (e.g., sensitivity or perturbation analysis) or against alternative approaches or untested conditions, or indeed used as a tool to test and answer a hypothesis-driven scientific question regarding neural control of human locomotion. I encourage the authors to address the above concerns, or at the minimum, clarify the (limited) scope and contribution of the current work, in the light of current knowledge and existing literature.

<minor comments="">

1. I cannot agree, or at least cannot understand, the statement “limited computational capacities in neural systems cannot handle the numerous parameter changes needed for each point-to-point velocity transition. (L39-41)”. Please elaborate or clarify.

2. If the model, and especially the reflex-based controller is essentially the same as what’s been provided in previous works, I suggest that the authors simply cite those work. At the minimum, some portions can be moved to the Supplementary materials. This way, the novel contribution of this work, which is the additional PWLS method used to implement velocity regulation (or control), can be also highlighted better.

3. Given the extensive and detailed methods, I request the authors use more organized/structured section headings for better readability, e.g., numbered rather than just by font sizes.

4. Presenting and discussing the results not only in terms of control parameters, costs (in objective function), and outputs, but also in terms of biomechanics (kinematics) and neural control (muscle activations) may also help.

5. What is the physiological basis for such a discretized, phase-dependent switching of control modes (L129-135)?

6. While CMA-ES is indeed suitable for global search in nonlinear and non-convex space, how would you validate the algorithm converged to a global minimum? More generally, do you have any insights as to what the landscape of the solution space looks like (e.g., redundant solutions with many local minima)?

7. The graphical presentation of the scheme for PWLS in the Supplementary material (video) was actually very helpful. If the authors intend to emphasize this, which is the novel contribution of this work, should they consider presenting this in the main manuscript?

Reviewer #3: This paper investigates the mechanisms of human walking, focusing on reflex-based control mechanisms and their impact on energy efficiency. To overcome challenges in velocity regulation, researchers developed a performance-weighted least squares (PWLS) method, which optimizes walking efficiency in a reflex-based bipedal system while maintaining target velocity. The study identified nine key reflex control circuits affecting energy efficiency at various walking speeds. The findings are expected to enhance the design of energy-efficient biped robots and walking assistive devices. However, this paper still leaves some major and minor questions unanswered.

Major:

The authors identify nine key control parameters for walking based on a 2D simulation model. However, the method of identification could be made more compelling. Please discuss whether these results are generalizable and uninfluenced by your preset model parameters (e.g., spring and damper constants of joints, sensory time delay, and weight coefficients in the objective cost function). Furthermore, could these findings be extended to a 3D context? As noted in the paper, such an extension could have practical applications.

Figure 7C indicates that for varying A, the values of many control parameters remain closely matched. When the optimized function of certain parameters is replaced with the one derived at A=800, the shift in energy efficiency is minimal (Fig. 9). I feel it is not convincing to deduce that these parameters are less critical.

Minor:

Line 58: Why is the chosen speed range only between 0.5 to 1.5 m/s? The typical human walking speed exceeds this range. Considering the emphasis on 'wide range' in the title, the chosen range appears somewhat restrictive.

Line 79: what is the joint torque function with spring and damper constants? Additionally, is the unit of the damper constant accurately represented? Shouldn't it be represented as Nms/rad instead?

Line 80: Could you elaborate on the reasons for the toe having distinct parameters and the rationale behind choosing the value 30?

Could you elucidate how the model depicts the interaction between the feet and the ground? Is there potential for sliding? If so, what is the assumed coefficient of friction?

Could you provide a more thorough explanation of equations 11-19 and 24-32? Where do they come from? What's the meaning of each term?

Line 251: A clear definition or criteria for 'steady walking' is required.

Fig. 7: Higher resolution or Scalable Vector Graphics are needed, as Fig. 7c becomes unclear when zoomed in.

Regarding Fig. 7C: The model falls when A = 100,000 for vtar_x = 1.2 and 1.5m/s. Why does the curve still include these data points? This raises questions about overfitting and its potential for extension.

**Have the authors made all data and (if applicable) computational code underlying the findings in their manuscript fully available?**

Reviewer #1: Yes

Reviewer #2: Yes

Reviewer #3: **No: **The provided GitHub repo lacks the data.

PLOS authors have the option to publish the peer review history of their article (what does this mean?). If published, this will include your full peer review and any attached files.

Reviewer #1: No

Reviewer #2: No

Reviewer #3: No

Figure Files:

Data Requirements:

Reproducibility:

</minor></major>

---

## [Decision Letter · Decision Letter 1]

13 Oct 2023

Dear Mr Koseki,

Thank you very much for submitting your manuscript "Identifying essential factors for energy-efficient walking control across a wide range of velocities in reflex-based musculoskeletal systems" for consideration at PLOS Computational Biology.

As with all papers reviewed by the journal, your manuscript was reviewed by members of the editorial board and by several independent reviewers. In light of the reviews (below this email), we would like to invite the resubmission of a significantly-revised version that addresses the concerns by reviewer #1.  

We cannot make any decision about publication until we have seen the revised manuscript and your response to the reviewers' comments. Your revised manuscript is also likely to be sent to reviewers for further evaluation.

Sincerely,

Jian Liu

Academic Editor

PLOS Computational Biology

James O'Dwyer

Section Editor

PLOS Computational Biology

Reviewer's Responses to Questions

**Comments to the Authors:**

Reviewer #1: The authors refer to the model of Wang as the basis for their analysis. Wang stated its purpose to be "Through comparisons to kinematic and torque data of human walking, we show that our results adopt a human-like torque generation strategy while producing kinematic data significantly closer to humans than previous work." (Wang 2012), which he arguably met (comp. fig. 6 in Wang 2012).

The authors of this paper seek to implement velocity control in the neuro-muscular model which has been done before, e.g. by Dzeladini in 2014 (10.3389/fnhum.2014.00371, cited by the authors), which in my opinion does not demonstrate problems with speed regulation, as claimed by the authors. The reviewer acknowledges the effort of the authors to address the reviewer's concern with their massive compilation of material. However, the cost of the authors' efforts to control velocity is the loss of human-like walking dynamics, as detailed below. This indicates issues with the formulation of the cost function and makes the results no less questionable than before.

The authors summary "[...] Overshoots and undershoots were observed in the measured GRF...[]" largely understates the dissimilarities of the gait. This leaves a model with a high number of parameters optimized for an unphysiological mode of locomotion which renders the insights questionable. Some of the references do not support the authors claims (see below). Some of the authors' added content raises more questions than it answers, e.g., how is the physical validity of the Mujoco caluclations etablished? To my knowledge the collision model in Mujoco is an approximation for faster computation but not an accurate physical contact model. The general question of my comment 3 remains - why did the author's choose the complex model of Wang instead of the simpler, original model of Geyer?

Specifically:

Fig. 8 Depicted GRFs and ankle angle trajectories substantially differ from human walking. Negative R-values for ankle range of motion are discussed with "At ankle joints, the cross-correlation values of ankle dorsiflexion were very low. The previous study also showed low cross-correlation values at ankle joints when optimized to minimize energetic cost [29]." (line 457 ff) - however [29] Song and Geyer 2019 do not show ankle range of motion cross-correlations in their study; in addition, "low" is very different from negative R-values Vertical GRFs show no double hump profile and several spikes in the initial phase of gait which should be addressed. Horizontal GRFs also show extreme peaks in the beginning which is neither feasible nor physiological.

Fig. 11 suggests that the A-value is a central property of the optimization, however the explanation of its meaning in line 276 ("Therefore, A determines the strength of the bias toward higher-performing data points.") is not helpful as is the lackluster explanation of PWLS in lines 280ff.

Side note: for M=125, 2M+1 is 251 not 250. However I do not understand what the sentence in l.274 is supposed to mean.

Reflecting on the authors conclusion, they failed to demonstrate the applicability of the optimized solutions and achieve parameter modulation while loosing fundamental properties that characterize walking. This largely reduces the applicability of the gained insights. I would suggest to revisit the underlying model and try to reduce the number of parameters rather than extend them. First and foremost, the resulting gait should aim at resembling human walking on a kinetic and kinematic level in order to draw conclusions concerning adaptation mechanicsms.

Reviewer #3: Thank you for your diligent efforts in addressing the feedback. I believe it has made a positive difference in the overall quality of the manuscript. Based on the updates, I am satisfied with how you've addressed the initial concerns and feel the manuscript has been enhanced accordingly.

**Have the authors made all data and (if applicable) computational code underlying the findings in their manuscript fully available?**

Reviewer #1: Yes

Reviewer #3: Yes

PLOS authors have the option to publish the peer review history of their article (what does this mean?). If published, this will include your full peer review and any attached files.

Reviewer #1: No

Reviewer #3: No
---

## [Editor Report · Decision Letter 2]

18 Dec 2023

Dear Mr Koseki,

We are pleased to inform you that your manuscript 'Identifying essential factors for energy-efficient walking control across a wide range of velocities in reflex-based musculoskeletal systems' has been provisionally accepted for publication in PLOS Computational Biology.

Best regards,

Jian Liu

Academic Editor

PLOS Computational Biology

James O'Dwyer

Section Editor

PLOS Computational Biology

---

## [Editor Report · Acceptance letter]

3 Jan 2024

PCOMPBIOL-D-23-00582R2 

Identifying essential factors for energy-efficient walking control across a wide range of velocities in reflex-based musculoskeletal systems

Dear Dr Koseki,

I am pleased to inform you that your manuscript has been formally accepted for publication in PLOS Computational Biology. Your manuscript is now with our production department and you will be notified of the publication date in due course.

With kind regards,

Anita Estes
